# BENCHMARKING DELETION METRICS WITH THE PRINCIPLED EXPLANATIONS

## ABSTRACT

Insertion/deletion metrics and their variants have been extensively applied to evaluate attribution-based explanation methods. Such metrics measure the significance of features by observing changes in model predictions as features are incrementally inserted or deleted. Given the direct connection between the attribution values and model predictions that insertion/deletion metrics enable, they are commonly used as the decisive metrics for novel attribution methods. Such influential metrics for explanation methods should be handled with great scrutiny. However, contemporary research on insertion/deletion metrics falls short of a comprehensive analysis. To address this, we propose the TRAjectory importanCE (`TRACE`) framework, which achieves the best scores of the insertion/deletion metric. Our contribution includes two aspects: 1) `TRACE` stands as the principled explanation for explaining the influence of feature deletion on model predictions. We demonstrate that `TRACE` is guaranteed to achieve almost optimal results both theoretically and empirically. 2) Using `TRACE`, we benchmark insertion/deletion metrics across all possible settings and study critical problems such as the out-of-distribution (OOD) issue, and provide practical guidance on applying these metrics in practice.

## 1 INTRODUCTION & BACKGROUND

With the rapid increase in computational power, deep neural networks have achieved remarkable success in many domains. Despite their impressive performance, DNNs are often criticized for their black-box nature, especially in critical applications where understanding decision-making process is crucial. To address this opacity, the field of explainable artificial intelligence (XAI) has emerged and developed rapidly, with various explanation methods introduced (Arrieta et al., 2020). Among these, attribution methods stand out and are widely used due to their straightforwardness and intuitive visualizations (Adebayo et al., 2018; Leavitt & Morcos, 2020). Given an input of $d$ features, such as pixels, tokens, patches, attribution methods assign an attribution value to each feature, illustrating its "importance" to the output. Such approach offers a clear insight into feature relevance and allows humans to directly comprehend it as it aligns well with the principles of linear models.

While attribution methods often take similar forms, they can originate from various methodologies and objectives. Given the same input data and the same black-box prediction model, different attribution methods can produce vastly different explanations. This variability presents a challenge for both end-users and researchers in selecting the most appropriate explanation method (Kaur et al., 2020; Krishna et al., 2022). To address this issue, evaluation *metrics* for attribution methods have been introduce to evaluate different explanations and identify the most suitable explanation approach. These metrics generally fall into two main categories: alignment and performance. Alignment metrics, such as the pointing game (Zhang et al., 2018), inspect how explanations align with the prior knowledge of the *data*. It has been critiqued that such metrics are actually evaluating the plausibility to humans rather than reflecting actual model behaviours (Jacovi & Goldberg, 2020; Wang & Wang, 2022b). In contrast, performance metrics such as insertion/deletion emphasize the model performance, where input features are perturbed (deleted/inserted, etc.) progressively according to their attribution values. Then the AUCs of the resulting curve, which contrasts model predictions against the proportion of perturbed features, serve as the evaluation criterion for the attribution method. For instance, when the most relevant features are deleted first (denoted as **MoRF**), a low AUC is anticipated. Conversely, when the least relevant features are deleted First (**LeRF**),

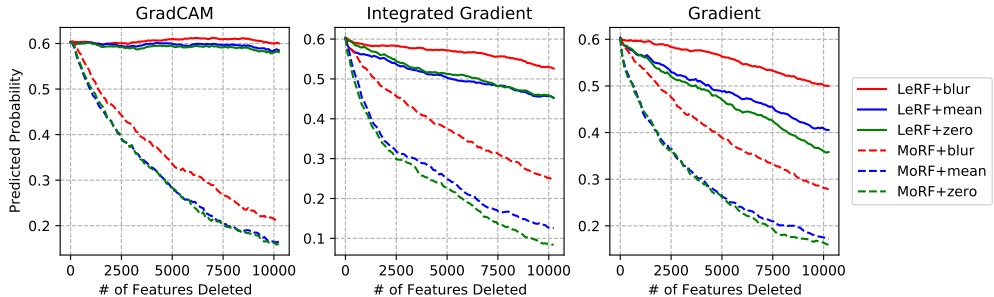

Figure 1: The deletion tests of GradCAM, Integrated Gradient and Gradient. Solid and dashed curves distinguish between LeRF and MoRF criteria. Different colors represent different reference types. We include the zero, mean and blurring references.

a high AUC is then expected. Deletion metrics characterize important features as those *affect the model prediction the most when progressively deleted*. The metric's widespread use suggests that such property is valued in the XAI community.

**Related Work of the Studies of Deletion Metrics.** Despite the deletion metrics' prominence as a preferred choice for evaluating attribution methods (Samek et al., 2016; Binder et al., 2016; Petsiuk et al., 2018; Chen et al., 2018; Qi et al., 2019; Schulz et al., 2020; Wang & Wang, 2021; Khorram et al., 2021; Covert et al., 2021; Chen et al., 2021), it is crucial to recognize that *metrics should undergo rigorous studies before widespread adoption*. Deletion metrics, with different settings such as the choices of the reference values that impute for the deleted features, LeRF/MoRF criteria, feature sizes, etc., may yield distinct results. Hence it is important to make a judicious choice among these variants in practice.

Besides, as the most paramount problem in the context of the deletion test, the OOD issue refers to the phenomenon that when only a small amount of input features are deleted, the input becomes out-of-distribution. As a result, the model performance decays significantly even if the informative features remain relatively intact. Although there are existing studies pointing out the OOD problem of the deletion metrics and proposing related workarounds (Hooker et al., 2019; Sturmfels et al., 2020; Schulz et al., 2020; Rong et al., 2022), they fall short in certain aspects. Hooker et al. (2019) propose to remove and retrain (ROAR) to alleviate the OOD issue. However, it requires training black-box models from scratch every time the number of deleted features changes, which is computationally expensive and hardly applied. Also, since the models change every time, it leans on explaining the dataset and the model family instead of the specific black-box model of interest (Sturmfels et al., 2020; Zhou et al., 2021; Ras et al., 2022). Schulz et al. (2020) argue that using MoRF or LeRF individually is insufficient and propose to use the difference between them as the measurement. But the statement lacks justifications. Rong et al. (2022) introduce remove and de-bias (ROAD), a weighted summation of the 8 surrounding pixels of the deleted one as the reference values, which is an intermediate stage between mean imputation and blurring. However, like other existing work, the proposed method is verified simply by observing whether the four selected explanation methods are ranked consistently under LeRF and MoRF. This raises risks because *studies of metrics should not be restricted by specific explanation methods*.

Tethered to popular explanation methods such as Gradient (Simonyan et al., 2013), Integrated Gradient (Sundararajan et al., 2017), GradCAM (Selvaraju et al., 2017) etc., existing studies of the deletion metric fall into a circular reasoning – These explanation methods, originally subjects of the deletion metric, are paradoxically used to validate the metric itself. Hence the assessment of the metrics will be highly biased by the selected explanation methods. For instance, to analyze the reference values in deletion metric, studies focusing on discrete attributions such as Gradient or IG are likely to conclude that the difference between reference values are significant, while studies focusing on smooth attributions such as GradCAM may conclude otherwise. Figure 1 shows the deletion tests of three methods, with zero, mean and blurring references. For Gradient and IG, different reference types lead to completely different scores. However, for GradCAM, the difference between zero reference and blurring reference is much less concerning. These opposite results suggest that studies of the metrics should not rely on specific explanation methods, but instead be approached in an *explanation-method-agnostic* fashion.

In response to these problems, we introduce the TRAjectory importanCE (TRACE) framework, which achieves the highest score of deletion metric both empirically and *theoretically*. By maximizing the score of the metric, TRACE is capable of (1) representing *what the metric really measures*, embodying the principled explanations associated with the deletion metric that reflect the exact influence of feature deletion on model predictions; and (2) benchmarking all the settings of the deletion metric, and providing guidance on how different choices can suffer from or be the remedy to the infamous OOD issue. The main contributions of this paper are summarized as follows.

- We formally study the mathematical essence of deletion metrics in an explanation-method-agnostic fashion to reveal intrinsic properties of the metrics.
- We propose TRACE, a combinatorial optimization framework to generate the *principled explanation* of the deletion metric and validate its near-optimality both empirically and theoretically. Thus it represents the exact feature importance under feature deletion.
- Using the principled explanation of the deletion metric, we present rigorous study on the various settings, and provide guidelines to effectively mitigate the OOD problem.

## 2 METHODOLOGY

In this section, the details of the TRACE framework are introduced. The discussion covers its solution using combinatorial optimization tools. And we introduce various settings of the deletion metrics and TRACE in Section 3. This section begins with a formalization of the deletion metric.

**Formalization of Deletion Metric.** Let $f : \mathbb{R}^d \to \mathbb{R}$ be a black-box model. An attribution method is defined as a mapping $\varphi_f : \mathbb{R}^d \to \mathbb{R}^d, \boldsymbol{x} \mapsto \psi$. For $\forall \delta \subseteq \{1, \cdots, d\}$, let $\boldsymbol{x}_{\backslash \delta}$ denote the input where features indexed by $\delta$ are deleted, and $\boldsymbol{x}_\delta$ denote it where features index by $\delta$ are kept. Then given the tuple $(f, \boldsymbol{x}, \psi)$, the deletion metric AUC score under the MoRF criterion can be written as

$$\text{MoRF}(\psi) = \sum_{k=0}^d f(\boldsymbol{x}_{\backslash \sigma(\psi)[k:]}) = \sum_{k=0}^d f(\boldsymbol{x}_{\sigma(\psi)[:k]}) \tag{1}$$

Similarly, $\text{LeRF}(\psi) = \sum_{k=0}^d f(\boldsymbol{x}_{\backslash \sigma(\psi)[:k]}) = \sum_{k=0}^d f(\boldsymbol{x}_{\sigma(\psi)[k:]})$. Here $\sigma$ maps the attribution map $\psi$ to a permutation of feature indices in the bottom-top order. That is, $\psi_{\sigma(\psi)[j]} \leq \psi_{\sigma(\psi)[j+1]}$. And $\boldsymbol{x}_{\backslash \sigma(\psi)[k:]}, \boldsymbol{x}_{\sigma(\psi)[:k]}$ represent the input data where (a) the last $k$ features indexed by $\sigma(\psi)$ are deleted and (b) the first $k$ features indexed by $\sigma(\psi)$ are kept, respectively. For example, if the attributions are $\psi = [0.1, 0.5, 0.3, 0.2]$, then $\sigma(\psi) = [1, 4, 3, 2]$. And $\boldsymbol{x}_{\sigma(\psi)[:1]} = \boldsymbol{x}_{\backslash \sigma(\psi)[3:]} = \boldsymbol{x}_{[1]} = \boldsymbol{x}_{\backslash[4,3,2]} = [x_1, \texttt{ref}_2, \texttt{ref}_3, \texttt{ref}_4]$ denote the input where the features $x_4, x_3, x_2$ are deleted. With the notations defined above, the best attribution-based explanation of the model prediction $f(\boldsymbol{x})$ under the deletion metric with MoRF criterion is naturally

$$\psi^*_{MoRF} = \arg \min_{\psi \in \mathbb{R}^d} \text{MoRF}(\psi) = \arg \min_{\psi \in \mathbb{R}^d} \sum_{k=0}^d f(\boldsymbol{x}_{\sigma(\psi)[:k]}) \tag{2}$$

Regrettably, the optimization of this objective to find the "best explanation" is infeasible. The study is confined to comparing the scores in Equation (1) between two attributions $\psi^1, \psi^2$. This limitation underscores why existing studies on the deletion metric rely heavily on specific explanation methods.

**Trajectory Importance (TRACE).** To address these challenges, we introduce TRACE. A crucial observation is that although attribution explanations are presented as dense vectors in the Euclidean space $\mathbb{R}^d$, their evaluations under the deletion metric are not based on the detailed attribution values. In fact, by defining that $\sigma(\psi)[i] < \sigma(\psi)[i+1]$ when $\psi_{\sigma[i]} = \psi_{\sigma[i+1]}$[1], an attribution $\psi$ can be mapped to a **unique** permutation of the indices $\{1, \cdots, d\}$. We thus define an equivalence relation $R$, where two attributions $\psi^1, \psi^2$ are equivalent when they map to the same permutation, i.e., $\psi^1 R \psi^2 \Leftrightarrow \sigma(\psi^1) = \sigma(\psi^2)$. And $\psi^1, \psi^2$ receive identical scores under the deletion metric. In consideration of this, we quotient out the equivalence class with the projection map $\mathbb{R}^d \to \mathbb{R}^d/R, \psi \mapsto [\psi]$. And since the equivalence class $[\psi]$ can be mapped to the permutation $\sigma(\psi)$ in a 1-to-1 manner, we have $\mathbb{R}^d/R \cong \mathcal{S}_d$. Here $\mathcal{S}_d$ denotes the symmetric group of order $d$, which consists of all permutations of $\{1, \cdots, d\}$. Proofs are shown in Appendix B.1. As a result, the original problem in Equation (2) transforms into an optimization over a finite, well-structured set $\mathcal{S}_d$ as

$$\text{TRACE-Mo: } \min_{\tau \in \mathcal{S}_d} \sum_{k=0}^d f(\boldsymbol{x}_{\tau[:k]}); \quad \text{TRACE-Le: } \min_{\tau \in \mathcal{S}_d} \sum_{k=0}^d f(\boldsymbol{x}_{\tau[k:]}) \tag{3}$$

---

[1] When the attributions are equal, features with smaller indices are put ahead.

To clearly differentiate between our framework and the test of the deletion metric under different criteria (e.g. MoRF, LeRF), we use the prefix TRACE (e.g. TRACE-Mo). In other words, MoRF is a evaluation criterion as defined in Equation (1), where lower values in MoRF indicates better explanations under the deletion metric; while TRACE-Mo is an optimization problem as we propose in Equation (3). This new formulation sets the stage for combinatorial optimization with adequate tools. We discuss the specific algorithms in Section 4.

**Trajectory to Attributions.** The optimizer of Equation (3) is a trajectory $\tau$ traversing all features in the bottom-top order. While the mapping from the attributions $\psi$ to the corresponding trajectory $\tau = \sigma(\psi)$ is surjective, $\tau$ can map back to attribution in its equivalence class. Define $\Pi_\tau = \{\pi | \pi : \mathcal{S}_d \to \mathbb{R}^d, \tau \mapsto \psi$, s.t. $\sigma(\psi) = \tau\}$ as the set of mappings from $\tau$ to attributions that preserve the trajectory. Thus $\forall \pi \in \Pi_\tau$, $\pi(\tau) \in \mathbb{R}^d$ is a valid attribution map of $\tau$. We define $\pi(\tau) = (\tau^{-1}/d)^\alpha \in [0,1]^d$, where $\tau^{-1} = \texttt{argsort}(\tau)$ is the ranking of features in the trajectory $\tau$ for simplicity. Here $\alpha$ controls the size of the highlighted region, which is similar to the colormap choices. Using $\pi$, we can map the optimization results $\tau$ from Equation (3) to attributions $\psi$, and visualize $\psi$ as heatmap, offering insights akin to attribution explanation methods. We visualize the TRACE results as heatmaps in Appendix C.

# 3 SETTINGS OF DELETION METRICS AND TRACE

As discussed in Section 1, various settings of the deletion metric give rise to a plethora of variants, resulting in distinct evaluation results even for the same $(f, \boldsymbol{x}, \psi)$ tuple, such as the differences shown in Figure 1. However, the judicious choice among these variants remains unclear. Here we discuss these possible variants comprehensively, and study how the choices of them can influence the metric via the principled explanations from TRACE in Section 5. Note that these settings influence both the metric through the term $f(\boldsymbol{x}_{\sigma(\psi)[:k]})$ in Equation (1) (i.e., when using deletion metrics for evaluation in practice), and the TRACE framework through the term $f(\boldsymbol{x}_{\tau[:k]})$ in Equation (3) (when determining the optimization objective). Also, it should be noted that the TRACE framework is compatible with any input data types. In this work, we focus on the image data, which is most influenced by the OOD issue.

**Deletion vs. Insertion.** Although the insertion metric serves as a popular alternative to the deletion metric and inserts features instead of deleting them, the differences between them are neutralized when the AUC is used for assessment. In fact, we prove in Appendix B.2 that they are equivalent and will focus on deletion in the following context for clarity.

**Theorem 3.1.** *The insertion metric is equivalent to the deletion metric up to AUCs with MoRF/LeRF.*

**Logit vs. Probability.** Model outputs, denoted by $f(\boldsymbol{x})$, vary in different contexts. For classifiers, both the predicted logit from the final linear layer and the probability yielded by the softmax activation can be seen as the output in standard practice. Notably, previous studies demonstrate that perturbations concerning logits differ from probabilities (Wang & Wang, 2022a). We include this variation with the suffixes -y (for logit) and -p (for probability).

**MoRF vs. LeRF.** The two criteria MoRF and LeRF, though sound symmetric, have very distinct interpretations. MoRF defines important features as *those who diminish the performance the most when deleted*. Conversely, LeRF sees features as crucial if they *maintain the performance the most when kept*. Taking both aspects into consideration, the "important features" should be able to diminish the model performance when deleted and preserve the model performance when kept. We denote this variant as LeRF−MoRF, which uses the difference $\sum_{k=0}^{d} \left( f(\boldsymbol{x}_{\tau[k:]}) - f(\boldsymbol{x}_{\tau[:k]}) \right)$ as the objective in Equation (3). In experiments, we consider all three variants: -Le, -Mo, and -Le−Mo.

**Reference Values.** Black-box models such as DNNs take inputs of a fixed size. Thus the deleted features have to be replaced with predefined reference values to represent the "null feature". The choices of reference values can significantly affect the results of the metric for some explanation methods. The current conventional way is to use heuristic methods such as zeros, means, and blurrings to avoid introducing exogenous information and overcomplicating problems (Lundberg & Lee, 2017; Sundararajan et al., 2017; Hooker et al., 2019; Shrikumar et al., 2017; Sturmfels et al., 2020; Covert et al., 2021; Rong et al., 2022). In fact, the choices of reference value types are tightly connected with the OOD issue via the trade-off between **deleting the feature** and **preserving the distribution**. The zero reference deletes features completely but breaks the input distribution severely.

In contrast, in the context of blurring reference, the original distribution is always preserved. However, the deleted features are also partially recovered, which can lead to problematic deletion test in practice. To study such influence, we include three types of reference values in our experiments: zeros, means, and blurrings.

**Input Feature Size.** Within an input image of size $224 \times 224$, the semantic meaning of pixel-wise attributions is very limited (Rieger et al., 2020). Grouping pixels and dealing with the superpixel patches, on the other hand, have been demonstrated to achieve great success (Dosovitskiy et al., 2020; Tolstikhin et al., 2021; Yu et al., 2022). It is also observed that the deletion metrics have been implemented with different resolutions. As a result, we operate on $t$ superpixel square patches, where the patch sizes are $\frac{224}{\sqrt{t}} \times \frac{224}{\sqrt{t}}$ (specially, when $t = 224 \times 224$, each patch is a pixel). By comparing the results of different patch sizes, we observe that the OOD issue is greatly mitigated by decreasing the resolution $t$ of the deletion process. Larger patches result in less noisy trajectories, but coarser explanations, while smaller patches lead to finer results but are much more vulnerable to the OOD problem. We study the influence of different patch sizes comprehensively in Section 5.2. And we will abuse the notations a little to denote by $\boldsymbol{x}_{\setminus \tau[k:]}$ or $\boldsymbol{x}_{\tau[:t-k]}$ the input image with the top $k$ patches deleted (i.e. bottom $t - k$ patches kept).

## 4 ALGORITHMS FOR TRACE

**Complexity Analysis.** The TRACE framework Equation (3) aim at finding a trajectory $\tau$ of features that optimize the "cost" defined by $(f, \boldsymbol{x})$. Therefore, it is a non-trivial problem and can be solved by combinatorial optimization with meta-heuristic algorithms. In Appendix B.3, we prove that TRACE is NP-hard by relating to the traveling salesman problem (TSP).

**Theorem 4.1.** *The optimization problem TRACE-Mo ($\{\min_\tau \sum_{k=0}^d f(\boldsymbol{x}_{\tau[:k]})\}$) is NP-hard.*

**Heuristic Approaches.** In order to quickly identify a trajectory of features that optimizes the objective outline Equation (3), one direct method is the greedy strategy. Instead of seeking an entire trajectory $\tau$ dynamically, this approach sequentially deletes one feature in each step. Starting from the highest ranked feature (the lowest one for TRACE-Le), it finds the feature that minimizes the prediction when deleted in each step. Such approach, while fast, is usually sub-optimal. Yet it reaches the global optimal of TRACE-Mo/Le if the features' contributions are additive, such as with linear models. It's essential to note, however, that this approach yields distinct trajectories for MoRF and LeRF, and thereby does not apply to the LeRF$-$MoRF test, resulting in ineluctable trade-offs between the principled explanation's optimality and efficiency. We demonstrate in Section 5 that TRACE-Greedy still outperforms all existing explanation methods significantly, and thus also serves the role as near-principled explanations w.r.t. feature deletion.

**Meta-Heuristic Approaches.** When benchmarking the deletion metric, the above compromise can cause insufficiency. Addressing the limitation described requires that the entire trajectory $\tau$ be optimized comprehensively. In such contexts, meta-heuristic algorithms are the judicious choice given their established efficacy in combinatorial optimization challenges (Baghel et al., 2012). Among them, simulated annealing (SA) (Kirkpatrick et al., 1983) has been actively employed in problems such as TSP to deliver sufficiently good sub-optimal results (Geng et al., 2011). Given its efficacy and the theoretical grounding, we too adopt SA in our methodology. The associated pseudo-code is provided in Appendix D. We also explore alternative meta-heuristic algorithms in Appendix E. In the following context, TRACE refers to TRACE-SA unless otherwise claimed.

**Neighbor Sets of SA.** The performance of SA depend on the apt choice of neighbors, especially on a discrete feasible set where the distance is not well-defined. Meanwhile, TRACE is essentially a harder problem than TSP, where the pairs of directly connected cities determine the total cost. TRACE considers not only the consecutively deleted patches but also the overall ordering of deleting patches matter. For instance, if a segment in the trajectory is reversed, TSP's costs only change for the segment's two endpoints. However, in TRACE, all values post the segment's initial change. As such, common neighbor strategies for TSP like vertex insertion, block insertion, and block reverse (Geng et al., 2011) do not transition to TRACE directly. Our comprehensive study on suitable neighbors for TRACE can be found in Appendix B.4, where we conclude that the optimal neighbor set should comprises all trajectories derived from the initial trajectory, $\tau_0$, by swapping two distinct features: $N(\tau_0) = \{\tau | \exists i, j, i \neq j, \tau_0[i] = \tau[j], \tau_0[j] = \tau[i]\}$.

**Optimality of the Algorithms.** While TRACE is capable of demonstrating exceptional performance in the deletion metrics, it should be acknowledged that when employing meta-heuristic/heuristic al-

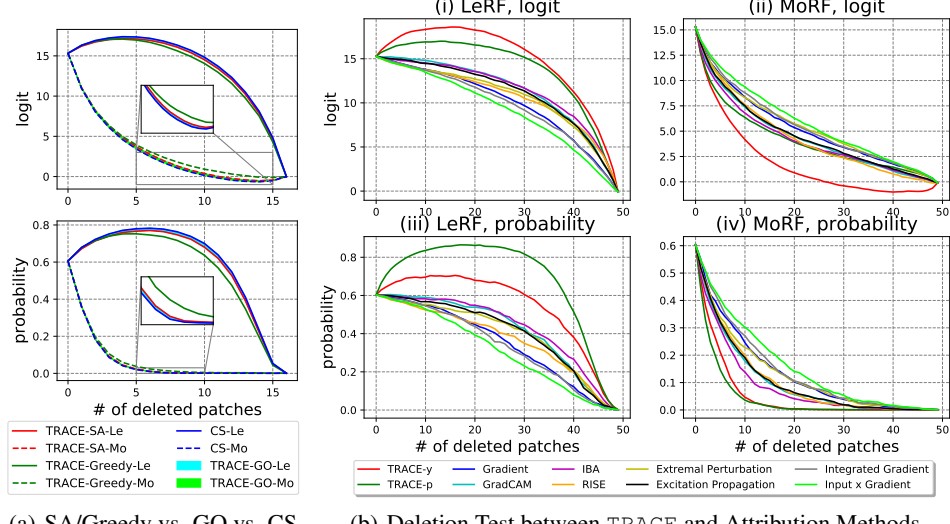

(a) SA/Greedy vs. GO vs. CS    (b) Deletion Test between `TRACE` and Attribution Methods

Figure 2: (a) The comparison between `TRACE`-SA/Greedy, `TRACE`-GO and Complete Search (CS). CS-Mo/CS-Le are the lower/upper bounds of `TRACE`-Mo/`TRACE`-Le, respectively. The blue, red and green curves are the results of CS, `TRACE`-SA and `TRACE`-Greedy. Solid and dashed curves are -Le and -Mo. The optimum `TRACE`-GO-Le and `TRACE`-GO-Mo lie in the cyan and lime color areas, respectively, which are notably marginal. (b) Deletion results of the first 200 images from the validation set of ILSVRC2012. In (i)(iii), patches are deleted following LeRF, and in (ii)(iv), patches are deleted following MoRF. The $y$-axis of (i)(ii) is the output logits of the network, and the $y$-axis of (iii)(iv) is the predicted probability. $x$-axis is the number of deleted patches.

gorithms, the resultant $\tau$ is not necessarily the global optimum. To validate the approximation to the optimum, we undertake an empirical study to bound the deviation between `TRACE`-SA/`TRACE`-Greedy and their global optimum. However, as a black-box optimization problem, the theoretical global optimum of Equation (3) (denoted by `TRACE`-GO) is inaccessible. Exhaustively searching for the global optimum is also impractical since $|S_t| = t!$. Therefore, instead of directly comparing with `TRACE`-GO, we propose complete search (CS), which is proved to be the lower bound of `TRACE`-GO. Formally, for $k = 1, \cdots, t$, CS-Mo solves for an index set $\boldsymbol{s}_k$ consisting of $k$ deletion features that minimizes the prediction $f(\boldsymbol{x}_{\backslash \boldsymbol{s}_k})$. Therefore, it is the lower bound of the corresponding term of `TRACE`-Mo in Equation (3): $\forall \tau \in \mathcal{S}_t, \forall k \in \{1, \cdots, t\}, f(\boldsymbol{x}_{\tau[:t-k]}) \geq \min_{\boldsymbol{s}_k \subset \{1, \cdots, t\}, |\boldsymbol{s}_k|=k} f(\boldsymbol{x}_{\backslash \boldsymbol{s}_k})$. And the equality will not hold unless $\forall k \in \{1, \cdots, t-1\}$, the optimizers $\boldsymbol{s}_k^*$ satisfy $\boldsymbol{s}_k^* \subset \boldsymbol{s}_{k+1}^*$. As a consequence, by summing up over $k$, we have

$$\text{TRACE-(Greedy/SA)-Mo} \geq \text{TRACE-GO-Mo} \geq \text{CS-Mo} \qquad (4)$$

Similar inequality holds for the -Le variant: `TRACE`-(Greedy/SA)-Le $\leq$ `TRACE`-GO-Le $\leq$ CS-Le. Therefore, by squeezing `TRACE`-(Greedy/SA) and CS, we can then verify the near-optimality of the algorithms (i.e., `TRACE`-(Greedy/SA) is close to the theoretical global optimum `TRACE`-GO).

## 5 EXPERIMENTS

In this section, we conduct experiments to 1) Validate `TRACE`'s optimality and the capability of serving as the principled explanation; and 2) Use `TRACE` to assess the impact of different settings (as discussed in Section 3) to address the OOD concern in deletion metric. We use a ResNet-18 model (He et al., 2016) as the black box $f$ for the demonstration. Other popular models such as AlexNet (Krizhevsky et al., 2017), VGG-16 (Simonyan & Zisserman, 2014), GoogLeNet (Szegedy et al., 2015), DenseNet-161 (Huang et al., 2017), and MobileNetV3 (Howard et al., 2019) are evaluated, too. We adopt the pre-trained weights from `torchvision`. Experiments utilize the ImageNet-1k (ILSVRC2012) dataset (Deng et al., 2009) with images resized to $224 \times 224$. For SA, we use $K = 5000$ iterations, initial temperatures of $T_0 = 2$ for -y and $T_0 = 0.1$ for -p, and a cooling rate of $\eta = 0.999$. Experiments are carried out on Intel(R) Core(TM) i9-9960X CPU @ 3.10GHz with Quadro RTX 6000 GPUs.

Table 1: The comparison among commonly studied DNNs on ILSVRC2012 with three different reference values. The tested models are (i) ResNet-18, (ii) VGG-16, (iii) AlexNet, (iv) GoogLeNet, (v) MobileNetV3, and (vi) DenseNet-161. Here we present the difference between AUCs of the *probabilities* for LeRF and MoRF, so larger values are desired.

| Ref. | M. | T-y | T-p | Grad | GC | IBA | RISE | EP | EBP | IG | IxG |
|---|---|---|---|---|---|---|---|---|---|---|---|
| Zero | (i) | 24.98 | **31.69** | 11.52 | 16.24 | 15.92 | 14.52 | 13.41 | 15.39 | 10.28 | 8.21 |
| | (ii) | 25.80 | **31.25** | 14.03 | 16.28 | 18.77 | 17.07 | 15.63 | 16.36 | 13.71 | 10.86 |
| | (iii) | 15.40 | **21.83** | 7.05 | 7.55 | 8.13 | 7.43 | 7.63 | 7.55 | 6.64 | 5.86 |
| | (iv) | 23.70 | **28.13** | 11.31 | 14.31 | 14.06 | 12.53 | 11.98 | 13.67 | 10.29 | 8.72 |
| | (v) | 27.55 | **33.86** | 8.15 | 16.78 | 13.06 | 10.0 | 11.08 | 10.74 | 8.49 | 6.20 |
| | (vi) | 28.00 | **35.25** | 11.35 | 19.82 | 18.87 | 18.18 | 17.04 | 19.17 | 12.20 | 9.33 |
| Mean | (i) | 25.64 | **32.51** | 11.45 | 16.22 | 15.66 | 14.57 | 12.96 | 15.40 | 10.51 | 8.48 |
| | (ii) | 26.66 | **32.64** | 14.09 | 17.00 | 19.06 | 17.63 | 15.73 | 16.58 | 13.83 | 10.77 |
| | (iii) | 16.33 | **23.32** | 8.91 | 9.82 | 9.20 | 10.65 | 9.83 | 9.82 | 8.48 | 6.73 |
| | (iv) | 24.02 | **29.17** | 11.63 | 14.49 | 14.14 | 12.70 | 11.85 | 13.89 | 10.63 | 9.06 |
| | (v) | 27.25 | **34.26** | 8.25 | 17.07 | 13.64 | 10.82 | 11.69 | 11.45 | 8.36 | 5.99 |
| | (vi) | 29.02 | **36.19** | 11.26 | 20.06 | 18.80 | 18.06 | 17.70 | 19.48 | 12.15 | 8.96 |
| Blurring | (i) | 27.34 | **33.84** | 10.93 | 17.38 | 16.41 | 16.01 | 14.98 | 16.24 | 10.04 | 7.57 |
| | (ii) | 27.50 | **34.09** | 14.51 | 17.89 | 19.27 | 18.15 | 15.81 | 17.06 | 14.73 | 11.46 |
| | (iii) | 19.55 | **26.78** | 8.90 | 9.81 | 9.20 | 10.65 | 10.00 | 9.81 | 8.47 | 6.73 |
| | (iv) | 24.86 | **29.83** | 11.60 | 14.50 | 14.31 | 13.44 | 12.55 | 13.89 | 10.95 | 9.24 |
| | (v) | 24.74 | **31.57** | 9.43 | 15.63 | 14.09 | 10.22 | 12.71 | 12.97 | 9.81 | 7.56 |
| | (vi) | 29.69 | **36.87** | 10.93 | 19.20 | 18.11 | 17.95 | 17.28 | 18.63 | 11.55 | 8.07 |

## 5.1 THE OPTIMALITY OF TRACE

**SA/Greedy vs. GO vs. CS.** To validate the optimization through Greedy and SA achieves the principled explanation of deletion metric, we demonstrate their closeness to CS, and thus squeeze the possible range of TRACE-GO. We test both -Mo with MoRF and -Le with LeRF. Because of the complexity of CS, we let $t = 4 \times 4 = 16$ here. As shown in Figure 2 (a), TRACE-SA (red) constantly outperforms TRACE-Greedy (green), suggesting better performance from meta-heuristic algorithms. And the difference between TRACE-SA (red) and CS (blue) is almost negligible, resulting in extremely squeezed areas for TRACE-GO between them, as shown in the cyan and lime areas (which are almost invisible). This suggests that TRACE-SA almost achieves the global optimum, and is capable of serving as the principled explanation of the deletion metric. TRACE-Greedy can also be used as the near-principled explanation when it's acceptable to trade performance for efficiency.

**The Optimality over Explanation Methods.** Conventionally, we compare TRACE with existing explanation methods, to demonstrate the deviation of existing explanation methods from the principled one. The results are demonstrated in Figure 2(ii). We present this comparison using TRACE-SA-Le$-$Mo. And the resolution of both TRACE and the deletion metric is set to $t = 7 \times 7 = 49$. We elaborate on these choices in the next section, where we benchmark all settings of the deletion metric with TRACE. Our observation highlights that existing attribution methods significantly underperform compared to the principled explanation provided by TRACE. Prior to TRACE's introduction, one might speculate that IBA (purple) is approaching the best AUCs given its superiority to other explanation methods. However, as shown in Figure 2(b) (i)(iii), TRACE reveals that the model performances can even increase substantially when unimportant features are deleted. In Figure 2(b), we use zero references for the demonstration. Further, we show AUCs across different reference values and black-box models in Table 1, where probability is used as the measurement. Same experiments for the logit can be found in Appendix G.

## 5.2 BENCHMARKING DELETION METRICS WITH TRACE

**Probability vs. Logit.** Comparisons between TRACE-p and TRACE-y in Figure 2(b) and Table 1 suggests that the principled explanations w.r.t. probability and logit align compatibly. For instance, in Figure 2(b):(iii)-(iv) and Table 1 where the evaluation is based on *probabilities*, though both TRACE-p and TRACE-y surpass all attribution methods, their discrepancy is non-negligible. In other words, TRACE-p (as the principled explanation for deletion metric with *probabilities*) performs better than TRACE-y in the evaluation using *probabilities*. Similar results are observed in Figure 2(b):(i)-(ii) when evaluating with *logits*. As a result, in practice, one should be aware of the desired goal (probability or logit) of the evaluation and select correspondingly.

**Reference Values.** Recall that in Figure 1, where features are defined as pixels, different reference types can affect the deletion test scores of explanation methods that focus on discrete attributions sig-

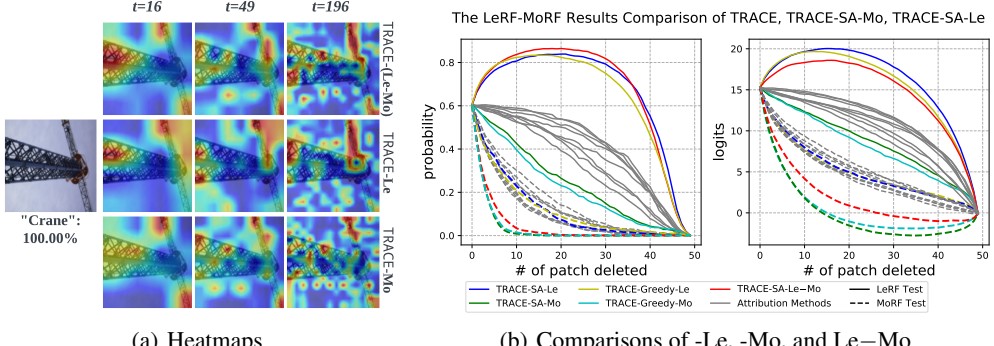

(a) Heatmaps          (b) Comparisons of -Le, -Mo, and Le−Mo

Figure 3: (a) The demonstration of the converted heatmaps of the image of a "crane" from the validation set of ILSVRC2012. A ResNet-18 predicts it correctly with the confidence $\approx 100.00\%$. The smooth factor $\alpha = 2$. We present results TRACE-SA-Le−Mo (top), TRACE-SA-Le (middle) and TRACE-SA-Mo (bottom) respectively. They are implemented w.r.t. the probability. And the we set $t = 4 \times 4$ (left), $t = 7 \times 7$ (middle) and $t = 14 \times 14$ (right). Here $t$ is the number of square patches of pixels for one image. (b) The comparison of the TRACE-SA-Le−Mo (red), TRACE-SA-Le (blue) and TRACE-SA-Mo (green), TRACE-Greedy-Le (yellow) and TRACE-Greedy-Mo (cyan) under the LeRF (solid) and MoRF (dashed) tests. All explanation methods are also included, but plotted indistinguishably just for the reference.

nificantly. However, as shown in Table 1, it can be found that the principled explanation TRACE (i.e., the highest-performing explanation) has consistent scores across different reference types.

**Patch Sizes.** In order to explain why the reference values no longer have that great influence, we explore how pixel sizes affect the OOD issue. From the heatmap of a crane image in Figure 3(a), we can observe that the principled explanation becomes more noisy as the patch becomes smaller (as $t$ increases from left to right), suggesting potentially more severe OOD problem.

For an impartial and rigorous verification, we execute a randomized deletion test in Figure 4, where different curves represent different patch sizes. Zero reference (left figure in Figure 4) deletes features completely, at the cost of pronounced OOD issue. And since patches are deleted completely at random, when the same amount of features are deleted, the difference in prediction decays among patch sizes is caused almost completely by the different OOD levels. And note Figure 4 reveals that smaller patches lead to a quicker decline in prediction quality. This suggests that *using larger patches effectively diminish the OOD issue.*

In contrast, blurring reference (middle figure in Figure 4) preserves the distribution, at the cost of not deleting the feature sufficiently. Thus although the decay of model prediction is slower, it might be caused by the information of the lingering features that should have been deleted instead of a mild OOD issue. Interestingly, as patch sizes increase, the difference between zero reference and blurring reference decreases (right figure in Figure 4). Recall that zero reference firmly deletes the features completely but compromising on the OOD issue, while blurring reference firmly solves the OOD issue but compromising on the feature deletion. Therefore, both desiderata can be attained when they behave the same – features are deleted, and the OOD issue is mitigated. This also explains the phenomenon in Table 1 where variances across different reference types are almost negligible.

**MoRF vs. LeRF.** MoRF defines important features as *those who affect the model prediction the most when deleted*. This, although seems symmetric to LeRF, is problematic. This is because the goal of MoRF is consistent with the OOD problem, where the deletion of a small amount of features can bring down the model prediction significantly. We demonstrate TRACE-Le−Mo, TRACE-Le and TRACE-Mo with different patch sizes in Figure 3(a) using heatmaps. Recall that smaller patches are likely to exacerbate the OOD issue, it can be found that the extent to which the methods are affected by OOD is ranked as TRACE-Le−Mo<TRACE-Le≪TRACE-Mo. We further verify this by a cross-validation between the principled explanations and the associated tests in Figure 3(b), where TRACE-(SA/Greedy)-Le is tested with MoRF and TRACE-(SA/Greedy)-Mo is tested with LeRF. As deduced, TRACE-Mo performs extremely poorly in the LeRF test, indicating that features recognized as "unimportant" by TRACE-Mo (i.e. deleted in the end) are not really unimportant.

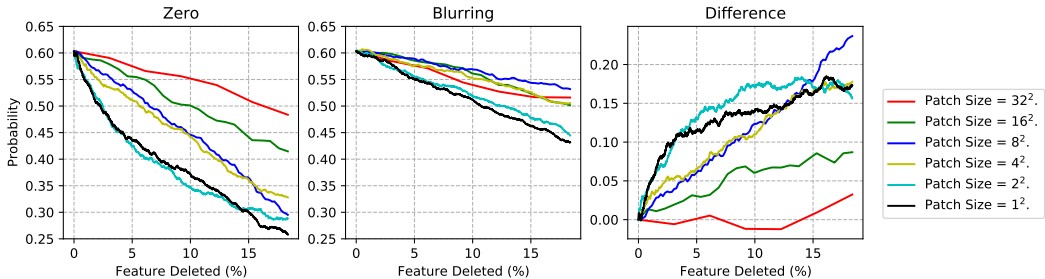

Figure 4: The random deletion test with (a) Zero reference (b) Gaussian blurring reference (c) Difference between (a)(b). 6 different patch sizes are tested.

Because when they are deleted first, the prediction drops fast (green, solid), too. On the other hand, those features that deleted in the end by TRACE-Le (i.e. important) do cause the prediction to drop fast when they are deleted first (blur, dashed). This result impartially benchmarks that LeRF should be the preferred criterion, while MoRF should be considered with great care. As the combined version, TRACE-Le−Mo compromises slightly under each criterion, but demonstrates perfect consistency. Therefore, *in practice, when choosing the criterion in deletion metrics, we suggest that LeRF−MoRF>LeRF≫MoRF.* As a complement, It is also interesting to notice that as $t$ decreases (the patch size increases) to $t = 16$, even TRACE-Mo is no longer affected. This is consistently supports the previous discussion of the patch size and the OOD issue.

**TRACE-Greedy as the Baseline.** The difference between TRACE-SA and TRACE-Greedy can be small according to Figure 3 (b). This illustrates that when associating with the MoRF and LeRF tests individually, the greedy scheme is an acceptable compromise to the meta-heuristic algorithms. As discussed above, both TRACE-(SA/Greedy)-Le can outperform all attribution methods in the LeRF−MoRF by a significant margin. And hence TRACE-Greedy can be used as a compromise between performance and efficiency. Furthermore, TRACE-Greedy-Le can also be used as an initialization of TRACE-SA to improve the speed of convergence. We provide assessment to the trade-off between performance and efficiency in Appendix F.

## 6 CONCLUSIONS

In this paper, we study the deletion/insertion metric, the most popular metric for the evaluation of attribution methods. We propose an explanation-method-agnostic framework TRACE that solves for the near-optimal deletion trajectories that approach the theoretical global minimum closely. In doing so, TRACE not only emerges as the principled explanation for the deletion metric, but also provides a standardized lens to inspect and benchmark all kinds of variants of the deletion metric. Our rigorous study offers several insights into the effective application of the deletion metric: (i) The image features should be deleted as superpixel patches instead of pixels. (ii) While MoRF and LeRF tests seem symmetric, the comparison between TRACE-Mo and TRACE-Le reveals that LeRF is a preferred criterion than MoRF. And LeRF−MoRF retains both sides to characterize important features. (iii) It is verified that, unlike pixel-wise deletion, the reference values' influence can be almost negligible when the features are deleted as superpixels. (iv) We also emphasize that using probabilities and logits yield distinct evaluation results, and thus the goal of the test should be explicit.

Furthermore, while TRACE is proposed as the intrinsic property of the deletion metric to study what such metrics are expecting, they are capable of mapping back to equivalent attributions that are unparalleled in the deletion metric – When the deletion metric is utilized, TRACE is **the one**. This phenomenon should be a warning that we should rethink how we develop and evaluate explanation methods. Since every time a metric is employed, there's a potential explanation-method-agnostic principled explanation for that metric. And the question remains, "is that the desired explanation?"

As we conclude, this work also leaves many interesting topics. For example, the deletion trajectories are closely related to path methods, where path methods use the incremental/decayed values as the attributions while TRACE considers the deletion/insertion order. Such differences might position them as distinct concepts, even with the same deletion trajectories. The exploration thus continues.

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
