## A    Extensive Related Work

**Attribution Methods.** In order to explain DNNs, numerous attribution methods have been developed. Based on the ways explanations are generated, they can be roughly separated into propagation methods and perturbation methods. Propagation methods back-propagate gradients or modified/pseudo gradients in a top-down fashion. Saliency (Simonyan et al., 2013) makes use of the gradient of input as the attribution values. Guided back-propagation (Springenberg et al., 2014) modifies the behavior of ReLU layers in backpropagations. LRP (Bach et al., 2015) and DeepLift (Shrikumar et al., 2017) change the back-propagation rule to propagate attribution values layer-wise. Input × Gradient (Shrikumar et al., 2017) uses the Hadamard product between input and its gradient as attributions. Sundararajan et al. (2017) propose axioms for attribution methods and introduce Integrated Gradient, which is the line integral of the input gradient. Grad-CAM (Selvaraju et al., 2017) generalizes the class activation mapping to all CNNs through the gradient of the CNN activations. Perturbation methods, on the other hand, usually generate explanations by modifying the input data and observing the change in the output. LIME (Ribeiro et al., 2016) locally approximates the prediction with a simple surrogate model. Occlusion (Zeiler & Fergus, 2014) identifies the object locations by replacing different portions of images with gray squares. SHAP (Lundberg & Lee, 2017) utilizes the approximated Shapley values (Shapley, 1953) as attribution values. RISE (Petsiuk et al., 2018) defines attribution values based on many randomly sampled masks. IBA (Schulz et al., 2020) generates explanations via per-sample information bottleneck. I-GOS (Qi et al., 2019; Khorram et al., 2021) optimize small masks to maximally decrease prediction scores. Fong & Vedaldi (2017) similarly optimize a relaxed continuous mask with $L_1$ regularization so that the predictions of the masked inputs are minimized. Agarwal et al. (2021b) formulate attribution generation as a Markov Decision Process and use reinforcement learning to solve it. Generally, perturbation methods are model-agnostic, meaning that they do not require any information about the explained model. On the contrary, propagation methods need access to the models (layers, parameters, etc.) to perform the propagation. There are also self-interpretable models with attribution values (Chen et al., 2019; Agarwal et al., 2021a; Wang & Wang, 2021; Li et al., 2021), where instead of explaining an existing black-box model, they propose entire new models that generate explanations and predictions at the same time.

**Insertion/Deletion Metrics.** As the most popular genre of evaluation metrics, insertion/deletion metrics are also called "faithfulness" in other works. They have a lot of variants, which, although have different names, all share the same essence. Samek et al. (2016) propose pixel flipping, where pixels are gradually replaced with zero values. This vanilla form is equivalent to most applications of such kinds of metrics (with names like ablations, maskings, etc.). Petsiuk et al. (2018) introduce the insertion metric in addition to the deletion one, where features are gradually inserted instead of deleted. Tomsett et al. (2020) carry out a sanity check and explore the AUC scores of multiple attribution explanation methods such as SHAP, Input × Gradient. Hooker et al. (2019) argue that deleting features from the input tends to break the original distribution. They propose ROAR to alleviate this issue. However, it requires training black-box models from scratch every time the number of deleted features changes, which is computationally expensive and hardly applied. In order to alleviate the out-of-distribution issue, there are other workarounds such as replacing feature values with reference values from mean/median/blurring instead of zeros (Wang & Wang, 2022a). Rong et al. (2022) propose to use the weighted summation of the 8 surrounding pixels of the deleted one as the reference values, which is actually an intermediate stage between mean and blurring. Also, for image data, it has been shown that deleting tiles of square pixels can also alleviate such issue (Schulz et al., 2020; Agarwal et al., 2021b). Schulz et al. (2020) also argue that removing features either from the top-down manner or the down-top manner individually is insufficient. They propose to use the difference between them as the measurement.

## B    Proofs & Analysis

### B.1    Details of the Formulation

In this section, we elaborate on the definitions of the equivalence relation $R$ defined over $\mathbb{R}^d$. The relation $R$ is defined so that every attribution map $\psi$ can be identified by its equivalence class $[\psi]$. However, note that when $\exists i, j \in \{1, \cdots, d\}, i \neq j$ such that $\psi_i = \psi_j$, the permutation $\sigma(\psi)$ is not well-defined. Therefore, we define that $\sigma(\psi)[i] < \sigma(\psi)[i + 1]$ when $\psi_{\sigma[i]} = \psi_{\sigma[i+1]}$, i.e., when

the attributions are equal, features with smaller indices are put ahead. And hence the relation $R$ is defined as follows.

**Definition B.1.** *We say the relation $R$ holds for $\psi^1, \psi^2 \in \mathbb{R}^d$ if they have they have the same permutation of features, i.e., $\psi^1 R \psi^2 \Leftrightarrow \sigma(\psi^1) = \sigma(\psi^2)$.*

Next, we prove the following theorem that the relation $R$ is an equivalence relation.

**Theorem B.2.** *$R$ is an equivalence relation.*

*Proof:* (i) $\forall \psi \in \mathbb{R}^d$, since an attribution map defines a unique permutation of features, we have $\sigma(\psi) = \sigma(\psi)$, and hence the reflexivity is proved by $\psi R \psi$. (ii) $\forall \psi^1, \psi^2 \in \mathbb{R}^d$, $\psi^1 R \psi^2 \Leftrightarrow \sigma(\psi^1) = \sigma(\psi^2) \Leftrightarrow \sigma(\psi^2) = \sigma(\psi^1) \Leftrightarrow \psi_2 R \psi_1$. Thus $R$ satisfies symmetry. (iii) For transitivity, $\forall \psi^1, \psi^2, \psi^3 \in \mathbb{R}^d$, if $\psi^1 R \psi^2$ and $\psi^2 R \psi^3$, then $\sigma(\psi^1) = \sigma(\psi^2) = \sigma(\psi^3)$. Therefore, $\psi^1 R \psi^3$ and the transitivity is proved.$\square$

Now that $R$ is an equivalence relation, we can effectively focus on the quotient set $\mathbb{R}^d/R = \{[\psi] : \psi \in \mathbb{R}^d\}$ that consists of all the equivalence classes instead of the original Euclidean space $\mathbb{R}^d$. Since the set $\mathbb{R}^d/R$ of equivalence class is not intuitive to deal with, we map the equivalence class $[\psi]$ to the permutation $\sigma(\psi)$ in a 1-to-1 manner, we have $\mathbb{R}^d/R \cong \mathcal{S}_d$. Here $\mathcal{S}_d$ denotes the set of all permutations of $\{1, \cdots, d\}$.

**Theorem B.3.** *$\mathbb{R}^d/R \cong \mathcal{S}_d$ with the bijection $[\psi] \mapsto \sigma(\psi)$.*

*Proof:* On the one hand, $\forall \tau \in \mathcal{S}_d$, let $\psi \in \mathbb{R}^d$ s.t. $\psi_{\tau[i]} = i$, then $\sigma(\psi) = \tau$. And thus $\exists [\psi] \in \mathbb{R}^d/R$ s.t. $[\psi] \mapsto \tau$. Hence it is surjective. On the other hand, $\forall \psi^1, \psi^2 \in \mathbb{R}$ s.t. $[\psi^1] \neq [\psi^2]$, by the definition of $R$, $\sigma(\psi^1) \neq \sigma(\psi^2)$. Hence $[\psi] \mapsto \sigma(\psi)$ is an injective. Therefore, $[\psi] \mapsto \mathcal{S}_d$ is bijective.$\square$

## B.2 PROOF OF THEOREM 3.1

Theorem 3.1. The insertion metric is equivalent to the deletion metric up to AUCs with MoRF/LeRF.

*Proof*: We show that deletion-MoRF is equivalent to insertion-LeRF. Deleting the most important $k$ features results in $\boldsymbol{x}_{\setminus \tau[k:]}$. On the other hand, inserting the least important $k$ features results in $\boldsymbol{x}_{\tau[:k]}$. Taking the summation of both of them, we have the equivalent AUCs:

$$\sum_{k=0}^{d} f(\boldsymbol{x}_{\setminus \tau[k:]}) = \sum_{k=0}^{d} f(\boldsymbol{x}_{\tau[:d-k]}) = \sum_{k=0}^{d} f(\boldsymbol{x}_{\tau[:k]}) \quad (5)$$

Thus it proves that deletion-MoRF is equivalent to insertion-LeRF. Similarly, it can be easily shown that deletion-LeRF is equivalent to insertion-MoRF. $\square$

## B.3 PROOF OF THEOREM 4.1

Theorem 4.1. The optimization problem TRACE-Mo ($\{\min_\tau \sum_{k=0}^{d} f(\boldsymbol{x}_{\tau[:k]})\}$) is NP-hard.

*Proof*: In TSP, a salesman traverses all $t$ cities, and the minimal cost is sought. It is defined by a cost matrix $\Delta = [\delta_{ij}]_{t \times t}$ where $\delta_{ij}$ is the cost going from city $i$ to $j$. Given a trajectory $\tau$, the cost function is defined as $f_{tsp}(\tau) = \sum_{i=1}^{t} \delta_{\tau[i]\tau[i+1]}$, where we extend $\tau[t+1] := \tau[1]$.

Note that $f(\boldsymbol{x}_{\tau[:k]}) = f(\texttt{ref})$ is constant w.r.t. $\tau$ when $k = 0$, it suffices to minimize $\sum_{k=1}^{t} f(\boldsymbol{x}_{\tau[:k]})$. Here we show this by demonstrating the corresponding decision problem "Given a cost $f^* \in \mathbb{R}$, is there a trajectory $\tau$ s.t. $\sum_{k=1}^{t} f(\boldsymbol{x}_{\tau[:k]}) \leq f^*$."

Now assume that there's a polynomial time algorithm for TRACE. Note that $f(\boldsymbol{x})$ is a black-box neural network and thereby can be any continuous function, and also $\forall i \neq j$ we have $\boldsymbol{x}_{\tau[:i]} \neq \boldsymbol{x}_{\tau[:j]}$, therefore, we define for any trajectory $\tau$ of length $t$ and $\forall i \in \mathbb{N}, 1 \leq i \leq t$,

$$f(\boldsymbol{x}_{\tau[:i]}) := \delta_{\tau[i]\tau[i+1]} \quad (6)$$

In this way for any trajectory $\tau$, we have

$$f_{tsp}(\tau) = \sum_{i=1}^{t-1} \delta_{\tau[i]\tau[i+1]} = \sum_{i=1}^{t} f(\boldsymbol{x}_{\tau[:i]}) \quad (7)$$

Therefore, this polynomial time algorithm also serves as an algorithm for TSP, a contradiction. $\square$

### B.4 NEIGHBOR SETS ANALYSIS

Note that $\tau$ can be any permutation of length $t$, which corresponds to $S_t$, the symmetric group of order $t$. Specifically, since $i = \tau[\tau^{-1}[i]] = \tau^{-1}[\tau[i]]$, we have $\forall \tau, \exists s \in S_t$ s.t.

$$s = \begin{pmatrix} 1 & 2 & \cdots & d \\ \tau^{-1}[1] & \tau^{-1}[2] & \cdots & \tau^{-1}[d] \end{pmatrix} = s(\tau), \tag{8}$$

which is a bijective. Since the feasible set $S_t$ is a discrete space, SA is modeled as a search method over a graph, where the vertices are feasible states, and the edges are possible movements between corresponding states, i.e. neighboring relations. Besides, it is also desired that each state has exactly the same number of neighbors. For the symmetric group $S_t$, such a graph is perfectly modeled by Cayley's graph (Magnus et al., 2004). Given a generating set $S \subset S_t$, the Cayley graph is defined as a directed graph $\mathrm{Cay}(S_t, S) = G(V, E)$ where the set of vertices $V$ are the same as $S_t$, and the arcs are defined by $E = \{[s_1, s_2] | \exists g \in S, \ gs_1 = s_2\}$, which results in an $|S|$-regular graph. Therefore, from any state $\forall s \in S_t$, we can move to $|S|$ other states. And there are also $|S|$ states that can move directly to $s$. For neighbors, we expect: 1) sufficiently small change between neighbored states and 2) the neighboring should be symmetric (i.e. $[s_1, s_2] \in E \Leftrightarrow [s_2, s_1] \in E$). Hence we only include transpositions (permutations that only exchange two elements) in $S$ (known as transposition set). For a transposition set $S$, we have $\forall s \in S, \ s = s^{-1}$, which means that $\mathrm{Cay}(S_t, S)$ is a symmetric directed graph and hence can be seen as undirected. In this case $G(\{1, \cdots, t\}, S)$ is known as the transposition graph, where the vertices are $\{1, \cdots, t\}$, and the edges are the transpositions in $S$. Then

**Proposition B.4.** *(Hahn & Sabidussi, 2013) $S$ generates $S_t$ if and only if $G(S)$ is connected.*

This indicates that $t - 1 \leq |S| \leq \frac{t(t-1)}{2}$, where the two equalities hold at spanning trees of the complete graph and the complete graph, respectively. Lakshmivarahan et al. (1993) propose several well-structured transposition generating set for $S_t$:

- Complete Transpositions: $S_{complete} = \{(i\ j) | 1 \leq i < j \leq d\}$
- Bubble-Sort Transpositions: $S_{bubble} = \{(i\ i+1) | 1 \leq i < t\}$
- Star Transpositions: $S_{star,i} = \{(i\ j) | 1 \leq j \leq d, j \neq i\}, 1 \leq i \leq t$

When applying SA over $S_t$, the number of states $t!$ is easy to explode compared with the neighbor size. This requires 1) sufficiently many movements from each state; 2) sufficiently few steps between any two states. In fact, let $\mathrm{diam}(G)$ denote the diameter of the graph $G$, then we have

**Theorem B.5.** $\mathrm{diam}(\mathrm{Cay}(S_t, S_{complete})) \leq t - 1$.

*Proof:* Given any two permutations of length $t$: $\forall \sigma_1, \sigma_2 \in S_t, \sigma_1 \neq \sigma_2$, we have where $t - 1$ transpositions are applied to $\sigma_1$. Note that $\forall i \in \mathbb{N}, i < t$, if $i = \sigma_1^{-1}[\sigma_2[i]]$, then the operation can be skipped. Therefore, there is always a path of length at most $t - 1$ connecting any two vertices in $\mathrm{Cay}(S_t, S_{complete})$. $\square$

On the other hand, for the bubble-sort transposition and star transposition, the diameters are (Akers & Krishnamurthy, 1989):

**Proposition 2.** $\mathrm{diam}(\mathrm{Cay}(S_t, S_{bubble})) = \frac{t(t-1)}{2}; \mathrm{diam}(\mathrm{Cay}(S_t, S_{star})) = (\lfloor 3(t-1) \rfloor)/2$

As a result, even though there are $t! = 49! \approx 6.08 \times 10^{62}$, the distance between any pair of vertices is only $t - 1 = 48$ in the complete graph. And this is the smallest value among all transposition sets. Because $S_{complete} = \cup_{S \text{ is a transposition set}} S$.

We present empirical results of different neighbor settings, including complete graph, bubble-sort graph, star-graph, vertex insertion (VI), block reverse (BR), block insertion (BI), and mix (89%BR + 10%VI + 1%BI) (Geng et al., 2011). The SA optimization process for the first 100 images of the validation set of ILSVRC2012 on pre-trained ResNet-18 provided by `torchvision` is plotted. The results are shown in Figure 5. It can be found that the complete graph outperforms other neighbor sets.

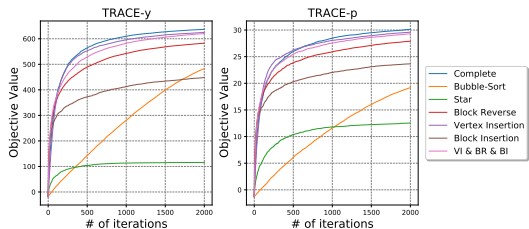

Figure 5: The comparison of different neighbor sets.

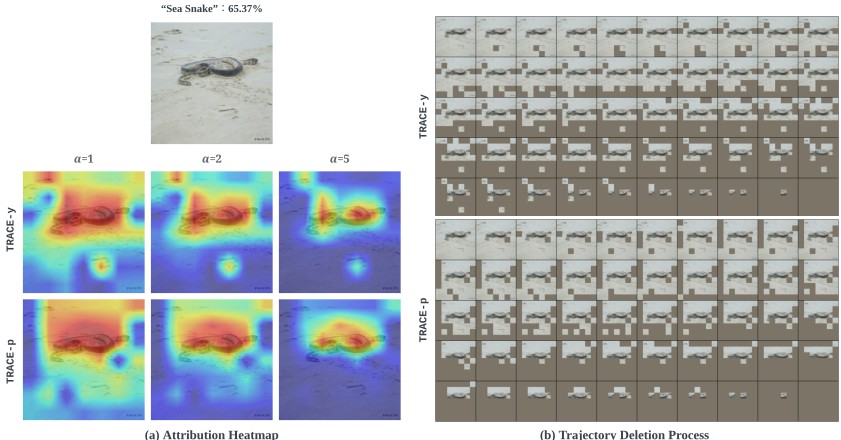

Figure 6: Visualizations of the optimized results of `TRACE-y` and `TRACE-p` on the "sea snake" image of ILSVRC2012 validation set. (a) The converted heatmaps $\psi$ with different smooth factor $\alpha$. (b) The deletion process is based on the trajectory $\tau$.

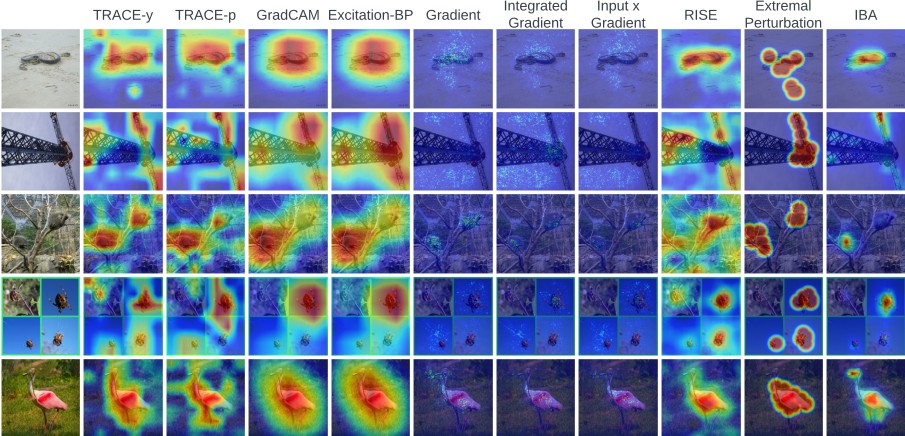

Figure 7: Visualizations of `TRACE` and popular attribution methods on images from ILSVRC2012.

## C VISUALIZING TRAJECTORIES AS HEATMAPS

In this section we visualize trajectory $\tau$ as heatmaps using $\psi = \pi(\tau) = ((\tau^{-1}/d)^\alpha) \in [0,1]^d$. Since in practice we implement `TRACE` in $t = 7 \times 7$ superpixel patches as discussed, we use the common practice in XAI, the bilinear upsampling, to interpolate the ranking of features back to the input space. Different choices of $\alpha$ are compared in Figure 6 (a). It can be found that $\alpha$ is independent

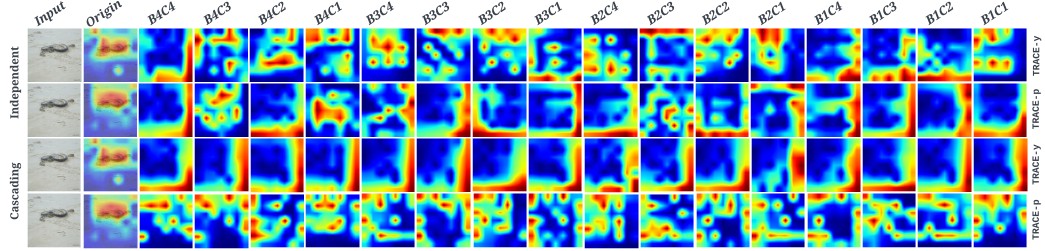

Figure 8: Sanity check using cascading randomization for TRACE. Convolutional layers of pre-trained ResNet-18 are randomized in the independent (upper) and cascading (lower) manners. In the independent randomization, other layers are kept at the pre-trained values. And in the cascading randomization, layers are progressively randomized from left to right (top-down). Here "B$a$C$b$" means the $b$-th convolutional layer in the $a$-th block.

from the deletion process shown in Figure 6 (b). Instead, it controls the visually highlighted area, serving similar purposes as colormaps in visualizations. In Figure 7, we demonstrate the deviation between existing attribution methods and the principled explanations of the deletion metric provided by TRACE.

As a convention test for attribution explanations, we also perform the sanity check for the converted heatmaps of TRACE. The results are shown in Figure 8, where the DNNs are randomized in either the independent or the cascading fashion. It is observed that TRACE passes the sanity check for explanation methods.

## D  PSUEDO-CODE FOR TRACE

---
**Algorithm 1** Simulated Annealing for TRACE

---
**Require:** black box $f$, input $x$, number of patches $t$, max iteration $K$, neighbor set
function `neighbor()`, initial temperature $T_0$, cooling rate $\eta$

> $T \leftarrow T_0$
> $\tau_0 \leftarrow$ `RandomInitialTrajectory`
> $auc_0 \leftarrow \sum_{k=1}^{t} \left( f(x_{\setminus \tau_0[:k]}) - f(x_{\tau_0[k:]}) \right)$
> $k \leftarrow 0$
> **while** $k < K$ **do**
> > $\tau_1 \leftarrow$ `RandomChoice(neightbor($\tau_0$))`
> > $auc_1 \leftarrow \sum_{k=1}^{t} \left( f(x_{\setminus \tau_1[:k]}) - f(x_{\tau_1[k:]}) \right)$
> > $\delta = auc_1 - auc_0$
> > **if** $\delta > 0$ **then**
> > > $\tau_0 \leftarrow \tau_1$
> > > $auc_0 \leftarrow auc_1$
> > **else**
> > > $r \leftarrow$ `RandomUniform`$(0, 1)$
> > > **if** $r < \exp(\delta/T)$ **then**
> > > > $\tau_0 \leftarrow \tau_1$
> > > > $auc_0 \leftarrow auc_1$
> > > **end if**
> > **end if**
> > $k \leftarrow k + 1$
> > $T \leftarrow \eta T$
> **end while**
> **return** $\tau_0$

---

---

**Algorithm 2** Greedy Scheme for `TRACE`

---

**Require:** black box $f$, input $\boldsymbol{x}$, number of patches $t$
   $k \leftarrow 0$
   $\tau \leftarrow$ `EmptyList`
   $\delta \leftarrow$ `[1,...,t]`
   **while** $k < t$ **do**
      $F \leftarrow$ `EmptyList`
      $N \leftarrow$ `EmptyList`
      $n \leftarrow 1$
      **while** $n <$ `len`$(\delta)$ **do**
         $\epsilon \leftarrow \tau \cup \{\delta[n]\}$
         $F \leftarrow F \cup \{f(\boldsymbol{x}_{\backslash \epsilon}\}$
         $N \leftarrow N \cup \{n\}$
         $n \leftarrow n + 1$
      **end while**
      $i \leftarrow$ `argmin`$(F)$
      $\tau \leftarrow \tau \cup \{\delta[N[i]]\}$
      $\delta \leftarrow \delta \backslash \{\tau[-1]\}$
      $k \leftarrow k + 1$
   **end while**
   **return** `flip`$(\tau)$

---

# E COMPARISONS AMONG DIFFERENT ALGORITHMS ON `TRACE`

As a supplementary, we test `TRACE` with several other popular algorithms for combinatorial optimizations. We include local search algorithms such as Hill Climbing, Tabu Search (GS) (Glover, 1986), and global search algorithms such as Genetic Algorithm (GA) (Holland, 1992). Note that these algorithms can have different complexity per iteration. Therefore, we compare the average optimization process within the same amount of time. As the benchmark, SA takes $\sim 200$ seconds for 5000 iterations when $t = 49$. Hence here we compare the results of these algorithms within 200 seconds, no matter how many iterations there are.

The results are shown in Figure 9, where Simulated Annealing outperforms other algorithms in the experiments. It should be noticed that one of the most important factors in `TRACE` is that the objective function is more expensive to evaluate than common combinatorial optimization problems like TSP. Thereby, an algorithm that fits `TRACE` well should require fewer evaluation times. For instance, Tabu Search requires evaluating all neighbors to update the tabu list, which means the complete graph cannot be applied as the neighbor size is $\frac{t(t-1)}{2} = 49 \times 48/2 = 1176$. The bubble-sort graph is applied instead, which is the reason why it is the slowest. This also corresponds to the results of the neighbor comparison experiments shown in Figure 5. Another interesting result is that Hill Climbing, which is not a meta-heuristic algorithm but a simple heuristic method instead, has the second-best result. This may suggest `TRACE` do not have many local optima in the feasible set $S_t$.

# F THE TRADE-OFF BETWEEN PERFORMANCE AND EFFICIENCY

The trade-off between the optimality of `TRACE` and the efficiency is inevitable given the nature of combinatorial optimization algorithms. And the running time is affected by the number of iterations.

When efficiency is preferred for explanations, `TRACE`-Greedy-Le or `TRACE`-SA-Le$-$Mo with a smaller number of iterations are preferred for explanations. When the ground truth is required for benchmarking of the metrics, we push the optimization process of `TRACE`to the limit by applying a larger number of iterations. We include the running time and the LeRF$-$MoRF deletion scores of all explanation methods and `TRACE`-Greedy, `TRACE`-SA. The results are shown in Table 2. Here Extremal Perturbation and RISE are implemented in the default settings suggested in the original papers. It can be found that (a) All TRACE variants outperform explanation methods in the deletion scores by a significant margin. (b) Increasing the number of iterations of SA consistently gives rise to the deletion score. (c) The running time of TRACE is very comparable and even outperforms popular perturbation-based explanation methods. With the trade-off on the efficiency, `TRACE` pushes

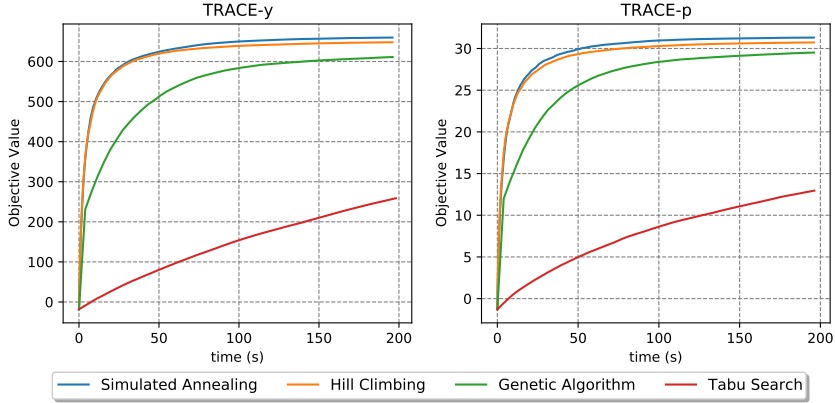

Figure 9: The comparison of different algorithms on solving `TRACE`.

Table 2: The comparison between the running time and the LeRF$-$MoRF

| Methods | LeRF-MoRF Score | Time (s) |
|---|---|---|
| Gradient | 11.52 | ∼0.005 |
| GradCAM | 16.24 | ∼0.005 |
| Excitation Back-Propagation | 15.38 | ∼0.014 |
| Integrated Gradient | 10.28 | ∼0.116 |
| Input $x$ Gradient | 8.21 | ∼0.006 |
| IBA | 15.92 | ∼0.144 |
| RISE | 14.52 | ∼5.161 |
| Extremal Perturbation | 13.41 | ∼28.975 |
| `TRACE`-Greedy-Le | **27.09** | ∼0.656 |
| `TRACE`-SA-Le$-$Mo ($K = 100$) | **28.42** | ∼4.079 |
| `TRACE`-SA-Le$-$Mo ($K = 500$) | **29.32** | ∼20.105 |
| `TRACE`-SA-Le$-$Mo ($K = 1000$) | **29.83** | ∼40.245 |
| `TRACE`-SA-Le$-$Mo ($K = 5000$) | **31.69** | ∼200.635 |

the deletion score to the limit gradually. (d) Back-propagation-based explanations do have the best efficiency.

## G    SUPPLEMENTARY RESULTS.

**Visualizations and Qualitative Inspections of `TRACE`.** Instead of attribution-based visualizations in the main manuscript, `TRACE` is better visualized as the deletion process. Because it is proposed that way. Here we demonstrate this in Figures 13 and 14, where the deletions are w.r.t. the probability and the logit respectively. The deletion process is visualized in the LeRF criterion. Hence the lastly deleted features are those that preserve the model's performance (or even significantly increase it as shown previously) when preserved. We focus on the deletion process following the probability in Figure 14.

In the first figure, we can see that the top left corner has a small bump on the ground, which is preserved till the last few steps. This indicates that keeping this bump can greatly preserve/increase the predicted probability of the snake class, suggesting that the model might recognize that feature as a part of snakes by mistake. Similarly, in the figure of row 3 column 2, the wings are the features that are preserved till the end, suggesting the importance of wings in recognizing birds. Also, it can be found in the figure of row 3 column 4 that the arms contribute greatly for the prediction of the crane.

**Exhaustive Results w.r.t. Logits.** Due to the space limit, we only present the exhaustive results w.r.t. the probability in Table 1. Here we include the results of the same experiment, where we

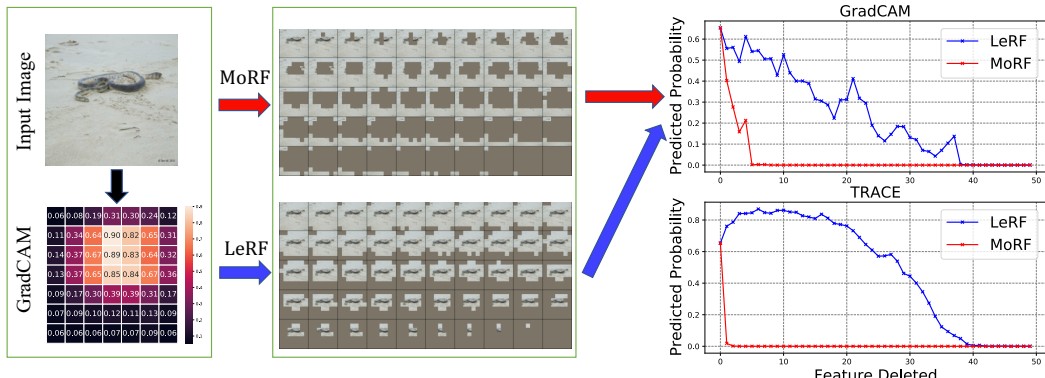

Figure 10: Demonstration of the deletion metrics under different criteria (MoRF or LeRF). The demonstration is based on ResNet-18 and GradCAM applied to the first image in ILSVRC2012. In the middle, MoRF (top) deletes the features with the highest attribution first, while LeRF (bottom) deletes the lowest feature first. On the top right, it shows the deletion metric results of MoRF (red curve) and LeRF (blue curve) for the single input using the probability as the indicators. On the bottom right, it shows the TRACE result for the same model and input.

focus on the logits instead. The results are shown in Table 3, and are consistent with the probability experiments.

**Extensive Comparisons of Attribution Methods.** In order to compare attribution methods comprehensively, we include more explanation methods such as guided GradCAM (Selvaraju et al., 2017), LRP (Bach et al., 2015), guided back-propagation (Springenberg et al., 2014), deconvolution (Zeiler & Fergus, 2014), GradientSHAP (Lundberg & Lee, 2017). The results are visualized in Figure 11. 13 methods apart from TRACE have been included in the test.

**Tabular Data.** Based on perturbing input features, the optimality of TRACE is independent of the data modality, making it readily applicable to other domains. Here we briefly demonstrate how TRACE works for tabular datasets. We train an MLP over the breast cancer dataset (Wolberg & Street, 1995), and test the explanation methods and TRACE using the deletion metric. Note that methods like GradCAM, Excitation-BP, IBA, etc. are specifically designed for image understanding. Therefore, here we only include explanation methods that are universal for all data types, including Gradient, Input × Gradient (IxG), Integrated Gradient (IG), Layerwise Relevance Propagation (LRP), and Gradient SHAP. The deletion results are shown in Figure 12. It can be found that TRACE (red) explores the principled deletion trajectory of features that push the limit of the deletion metric to a new level. Without the red curves, one can hardly imagine the seemingly good AUCs are still far from the optimum.

## H    TRANSFERABILITY OF THE OPTIMAL TRAJECOTIRES

Now that TRACE can be seen as a principled explanation of a model reflecting the influence of feature deletion, it can serve other uses. Previously, given $x, f_1, f_2$ and an explanation method $\varphi$, the explanation method provides explanations $\phi_{f_1}(x), \phi_{f_2}(x)$ for the two models, respectively. It can be observed that although the two explanations reflect the mechanism of the corresponding models, they do not provide an opportunity for the evaluation of the cross-model mechanism – e.g. the explanation $\phi_{f_1}(x)$ of model $f_1$ has very limited meaning for model $f_2$. Thus explanations are usually compared through $d(\varphi_{f_1}(x), \varphi_{f_2}(x))$ where $d(\cdot, \cdot)$ is usually a distance defined on $\mathbb{R}^d \times \mathbb{R}^d$. It is well-known such distance is limited due to the curse of dimensionality. Inspired by (Madry et al., 2017; Shi et al., 2019), we study the transferability of TRACE among different models. The correlation matrices are reported in Figure 15. All six matrices are the combinations of the two indicators: the logit and the probability, and the three references: zeros, means, and Gaussian blurs. By considering the transferability of the trajectory between the two models, we are able to quantify the difference between models in a more reasonable way.

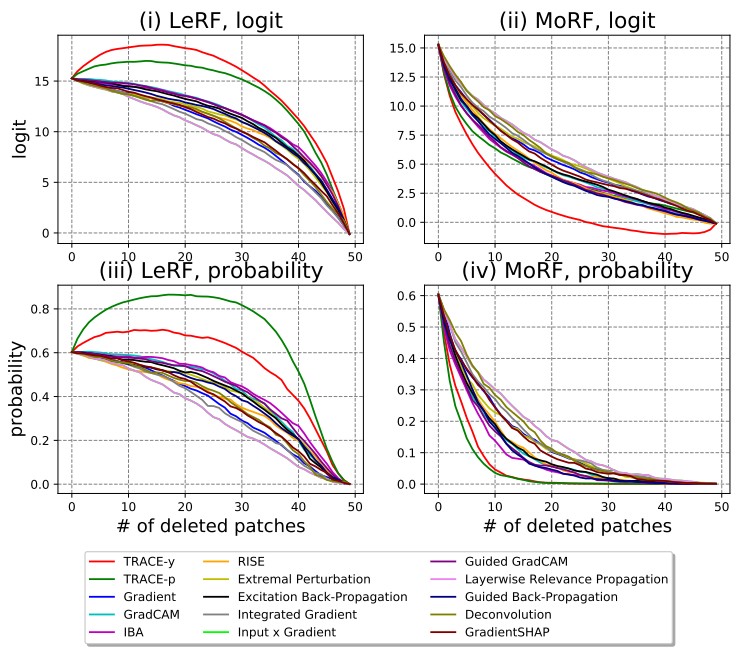

Figure 11: The comparison between the deletion tests of `TRACE` and more attribution methods

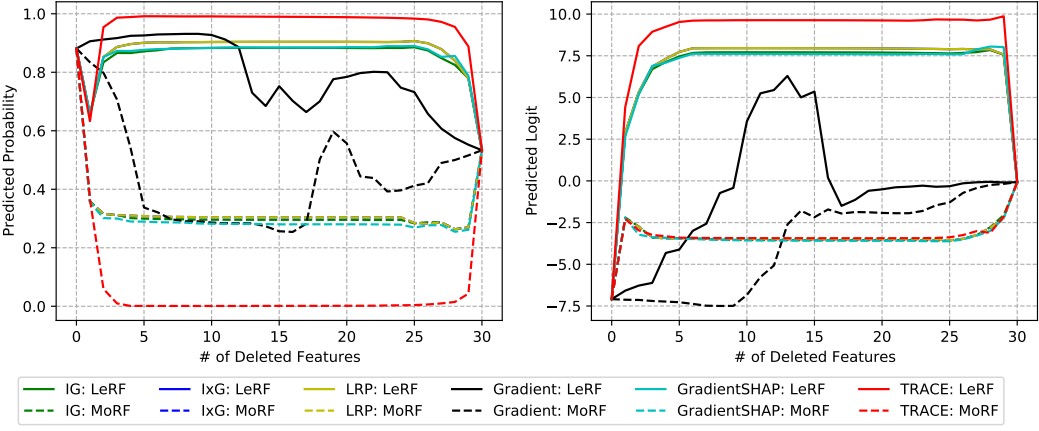

Figure 12: Deletion test results of the breast cancer dataset.

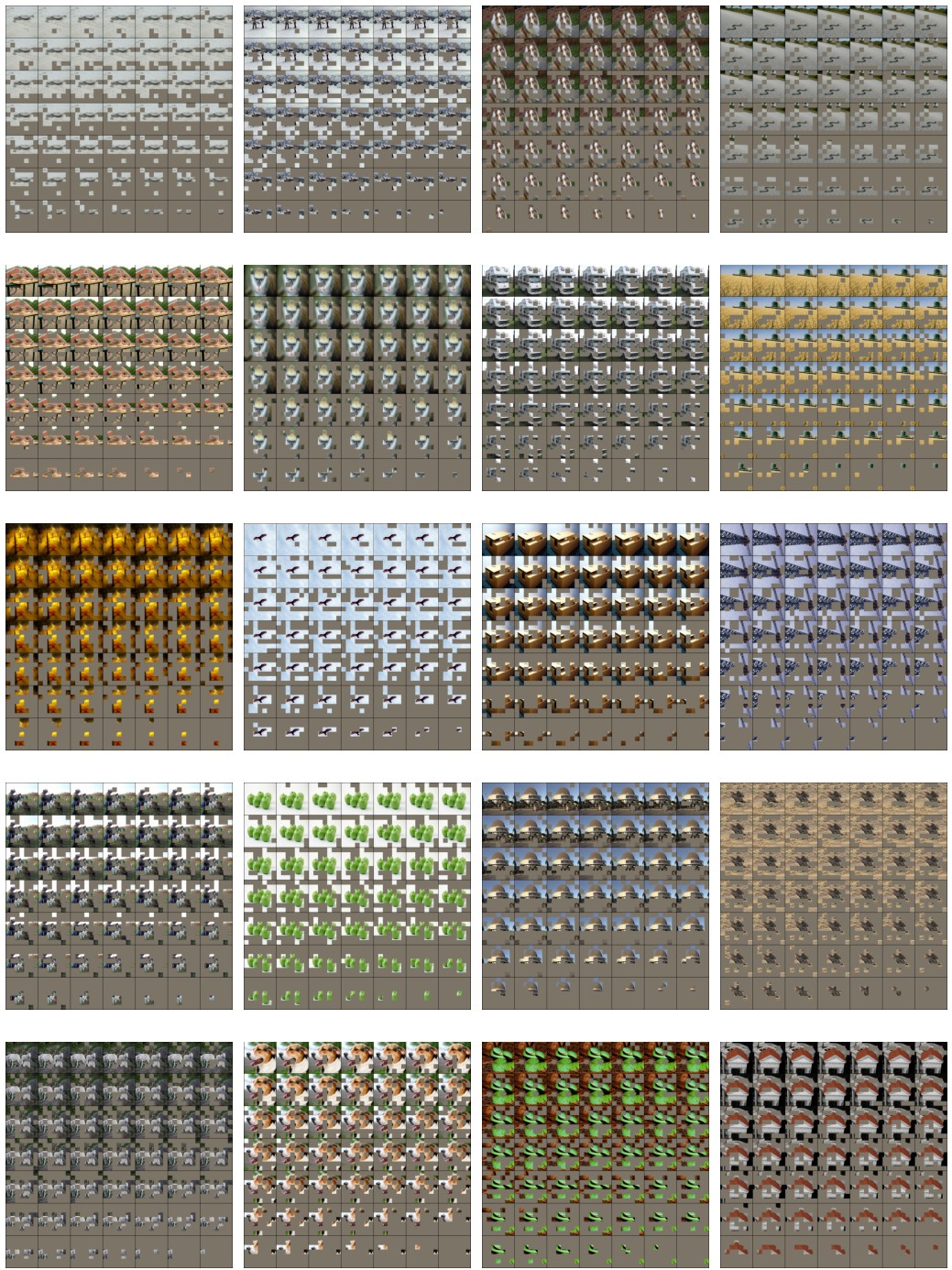

Figure 13: The deletion process of images from ILSVRC2012 on ResNet-18. Here all the trajectories are generated under TRACE-y. That is, the predicted logits are the measurement.

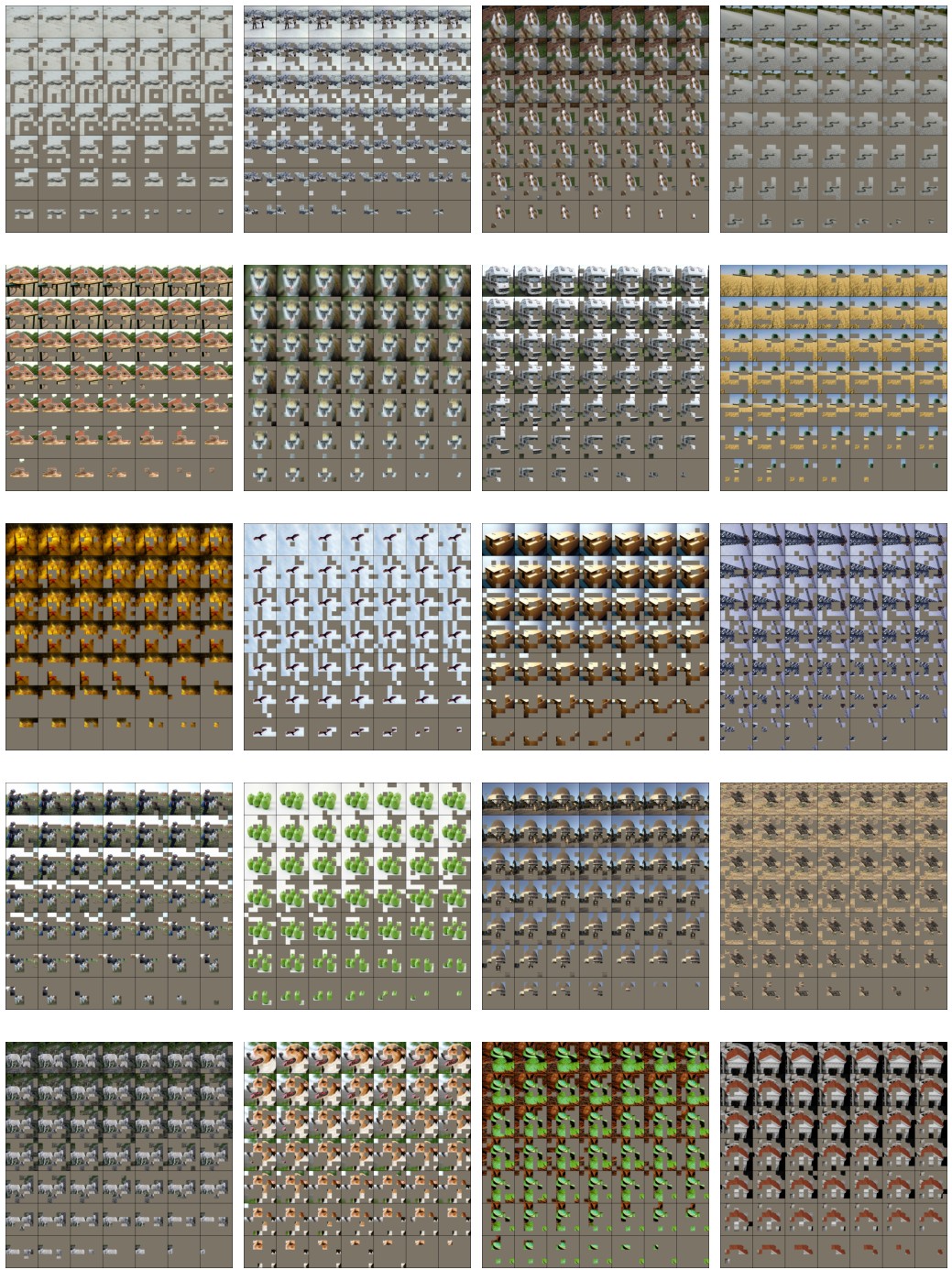

Figure 14: The deletion process of images from ILSVRC2012 on ResNet-18. Here all the trajectories are generated under TRACE-p. That is, the predicted probability is the measurement.

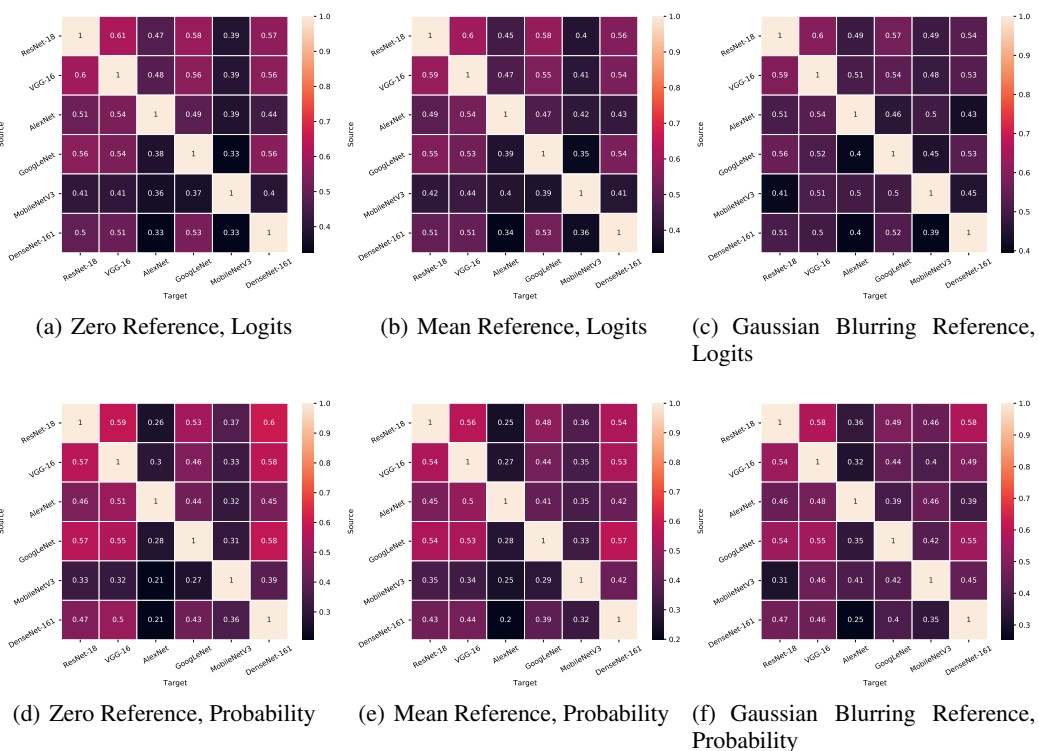

(a) Zero Reference, Logits

(b) Mean Reference, Logits

(c) Gaussian Blurring Reference, Logits

(d) Zero Reference, Probability

(e) Mean Reference, Probability

(f) Gaussian Blurring Reference, Probability

Figure 15: The transferability of the trajectory among 6 different DNNs. In each subfigure, the $y$-axis is the list of source models, which the trajectories are optimized over. The trajectories are then tested over the target models listed on the $x$-axis. The upper row is the result when the predicted logits are used as the measurement. The lower row is the result of the predicted probability. And each column represents a certain type of reference value. Results show that AlexNet and MobileNetV3's best trajectories are the most distant from others.

Table 3: The comparison among commonly studied DNNs on ILSVRC2012 with three different reference values. Here we present the difference between AUCs of the *logits* for LeRF and MoRF.

| Reference | Model | T-y | T-p | Grad | GC | IBA | RISE | EP | EBP | IG | IxG |
|---|---|---|---|---|---|---|---|---|---|---|---|
| Zero | ResNet-18 | **654.63** | 494.39 | 246.28 | 345.30 | 368.63 | 330.01 | 271.24 | 330.95 | 217.46 | 170.96 |
| | VGG-16 | **733.33** | 570.08 | 354.31 | 389.71 | 446.96 | 422.11 | 342.14 | 412.27 | 340.36 | 273.04 |
| | AlexNet | **528.92** | 387.99 | 235.47 | 215.60 | 234.70 | 262.67 | 225.70 | 215.60 | 228.04 | 199.65 |
| | GoogLeNet | **436.03** | 373.45 | 186.15 | 230.28 | 217.85 | 217.04 | 184.37 | 219.79 | 170.78 | 147.61 |
| | MobileNetV3 | **564.07** | 412.34 | 140.08 | 279.24 | 228.40 | 198.11 | 173.48 | 200.90 | 146.53 | 111.26 |
| | DenseNet-161 | **748.21** | 553.13 | 231.28 | 389.69 | 361.18 | 375.25 | 312.25 | 377.49 | 236.95 | 73.57 |
| Mean | ResNet-18 | **677.82** | 490.32 | 255.54 | 349.48 | 332.02 | 332.77 | 259.02 | 334.96 | 225.60 | 179.44 |
| | VGG-16 | **761.62** | 563.88 | 360.30 | 401.50 | 456.45 | 432.31 | 352.19 | 423.36 | 348.54 | 276.86 |
| | AlexNet | **541.18** | 396.74 | 252.16 | 242.20 | 241.08 | 313.22 | 257.51 | 242.20 | 235.24 | 197.67 |
| | GoogLeNet | **446.35** | 368.47 | 189.38 | 233.79 | 218.98 | 218.78 | 182.57 | 223.79 | 175.87 | 152.19 |
| | MobileNetV3 | **568.10** | 411.44 | 146.17 | 284.28 | 238.15 | 210.70 | 183.48 | 212.35 | 148.70 | 109.74 |
| | DenseNet-161 | **755.73** | 521.30 | 231.56 | 389.62 | 357.16 | 368.78 | 315.32 | 380.62 | 237.05 | 173.82 |
| Blurring | ResNet-18 | **667.50** | 488.25 | 237.36 | 347.15 | 329.25 | 338.98 | 270.65 | 332.68 | 218.25 | 169.08 |
| | VGG-16 | **785.37** | 572.89 | 355.51 | 409.08 | 457.24 | 450.51 | 356.16 | 420.88 | 350.53 | 279.70 |
| | AlexNet | **580.66** | 424.06 | 252.31 | 242.59 | 240.95 | 313.36 | 256.44 | 242.59 | 235.26 | 197.64 |
| | GoogLeNet | **441.68** | 369.25 | 178.26 | 218.80 | 206.72 | 221.96 | 180.74 | 207.93 | 169.69 | 144.81 |
| | MobileNetV3 | **567.15** | 419.61 | 180.00 | 296.19 | 264.50 | 221.05 | 227.01 | 251.26 | 177.09 | 135.86 |
| | DenseNet-161 | **724.62** | 507.72 | 194.39 | 340.50 | 317.74 | 337.16 | 286.58 | 331.22 | 199.23 | 141.73 |