# OpenReview forum: "Benchmarking Deletion Metrics with the Principled Explanations"
_ICLR.cc/2024/Conference — Submitted to ICLR 2024_

### Official Review · Reviewer_wwkN · 2023-10-25

**Soundness:** 3 good
**Presentation:** 3 good
**Contribution:** 3 good
**Rating:** 8
**Confidence:** 4

**Summary:**

The authors propose an attribution methods that maximizes the a deletion/insertion metric, namely the AUC of the logit or probability.
The authors show that this optimization problem can be stated as a combinatorics one that, although being NP-hard can be approximated using existing methods.
Unsurprisingly the method outperforms any baseline with respect to that metric since it optimizes it.
The paper contains extensive analyses especially on the difference between masking the most relevant features first or the least ones, what kind of masking, the use of logits or probabilities, etc.

**Strengths:**

* The paper tackles an important issue and proposes an elegant solution.
* The paper is well written an easy to follow.
* The paper contains extensive analyses of different aspects of the problem.

**Weaknesses:**

* It's a journal paper that is cut to be conference paper. Important points are in the supplementary that are often difficult to find and thus rarely read.
* Maybe I overlooked something, but the relevance of deletion/insertion methods and their conformity to what humans/experts expect is somehow missing. Maybe some qualitative results in the main paper would be nice.
* The proposed method obviously outperforms existing ones. I think a ranking of the baselines is missing.

**Questions:**

* Could you provide a ranking the baseline methods and an analysis thereof. Yes, TRACE outperforms. It has to since it optimizes the metric. Yet it would be interesting do use these analyses to learn something about other more widely used existing methods.
* Speaking of existing method, I believe LRP (which is a principled method leveraging the weights) is missing.
* Blurring is never clearly defined.
* Plots are not always clear, as many curves may overlap. Especially, Fig2a is unreadable as -GO curves are difficult to find, even when zoomed in. Maybe represent the difference, instead. The true value is not very interesting here.
* Table 1: are the experiments repeated? do you report means? This table is not very well discussed and it contains anyway too many information. I don't have a solution, I just point it out. But again , it bols down again to the fact that only TRACE is studied, although insights on other methods would be interesting, eg. standard deviation between architectures.
* The references seem to be in an arbitrary order. I am not sure if it is ICLR's template that is so, but an alphabetical order would be better.

* There are several typos or un-introduced abbreviations.
* Double check the references some are broken, eg. Wang and wang 2022a.
* p3: you defined $\psi=[.1,.5,.3,.2]$ and then use $\phi_f(x)$ this is confusing and clutter the text. I suggest, to avoid using $\phi_f at all$:
For example, if the attributions are $\phi_f(x)=\psi=[.1,.5,.3,.2]$ then $\sigma(\psi)=[1,4,3,2]... etc
* The first paragraph of page 4 is convoluted although the point is simple.
* p6 ... which is proved to be __a__ lower bound.. not [the] lower bound
* p6 typo: GO-MO solves for an index [s]$s_k$..

---

> ### Author Response · Authors · 2023-11-18
> **Response to Reviewer wwkN**
>
> **Response to Revewer wwkN**
>
> We appreciate the reviewer's acknowledgement of our work. We address the concerns as follows.
>
> **[Relevance of Deletion/Insertion Metrics and Their Conformity to What Humans expect]**
>
> Deletion metrics are the most popular evaluation metrics of attribution methods because of its conformity to humans intuition, while also generating the evaluation scores of explanations via interactions of the black-box models being explained. Specicially, an attribution explanation present straightforward scores of input features, representing the levels of relevance/importance/sensitivity of the features w.r.t. the black-box model's prediction. Therefore, it is natural to evaluate the quality of the explanations by inspecting "How would the black-box model perform without these explanations?" Furthermore, note that due to the correlations and nonlinearities of features, deleting $k$ features and deleting $k+1$ features do not lead to additive predictions of the black box. Therefore, the progressive deletion of features are preferred for it explores all possible $k$s.
>
> We also present additional qualitative analysis of TRACE -- the principled explanation of the deletion metrics, to demonstrate how progressive deletion works in finding features that affect the model's performance the most in the deletion scenario. The analysis can be found in **Appendix G**, where we carefully inspect the visualized deletion trajectories solved by TRACE.
>
>
> **[Rankings of Baselines]**
>
> The proposed TRACE framework aims at benchmarking the most popular deletion metric for explanation methods rather than the explanation methods themselves, and thus the analysis and the conclusions were not focused on specific existing explanation methods.
>
> Using the benchmarked variants of the deletion metrics, we can have the following rankings of existing explanations under the verified deletion metric. The rankings illustrate "*to what extend can the selected features of these explanation methods affect the model's behavior*":
> - Probability: GradCAM (GC) $>$ Information Bottleneck Attribution (IBA) $>$ RISE $>$ Excitation Back-Propagation (EBP) $>$ RISE $>$ Extremal Perturbation (EP) $>$ Gradient (Grad) $>$ Integrated Gradient (IG) $>$ Input $\times$ Gradient (IxG)
> - Logits: GradCAM (GC) $\approx$ RISE $>$ Information Bottleneck Attribution (IBA) $>$ Excitation Back-Propatation (EBP) $>$ Extremal Perturbation (EP) $>$ Gradient (Grad) $>$ Integrated Gradient (IG) $>$ Input $\times$ Gradient (IxG)
>
> We have the following remarks on the results
>
> (1) The results suggest that (a) GradCAM is the state-of-the-art among all existing attribution methods for selecting features that maintain the model's performance when kept and harm the model's performance when deleted; and (b)perturbation-based methods (eg. RISE, EP, etc.), although related more closely to feature perturbations, do not have significant superiority over back-propagation-based method (GradCAM, EBP, Gradient, etc.) under the deletion metric. In fact, (3) it is noticed that pixel-wise attributions (e.g. IG, IxG and Gradient) tend to underperform the deletion metrics. It is the feature size (pixel-level, patch-level, etc.) that affect more in the deletion metric rather than the difference between back-propagation and perturbation.
>
> (2) Therefore, we advocate that the metrics for explanations should serve specific and practical purposes. For instance, deletion metrics measure "to what extent can the selected features of these explanation methods affect the model's behaviour", Invariance/Equivariance Robustness [1] measures whether the highlighted features are invariant/equivariant w.r.t. transformations, etc.
>
> (3) By finding the optimal explanation of the deletion metrics, we also emphasize that the evaluation of existing explanation methods should not be based on a single and universal metric. Failing certain metrics does not instantly disprove explanation methods (e.g. gradient in deletion test). It's also likely that the metrics and the explanation methods simply focus on different goals.

---

> ### Author Response · Authors · 2023-11-18
> **Response to Reviewer wwkN**
>
> **[Layerwise Relevance Propagation (LRP)]**
>
> We thank the reviewer very much for mentioning the LRP methods. In fact, the rule-based back-propagation explanation method LRP [2] is closely related to gradient-based methods. Proposition 3 in [3] demonstrates that $\epsilon$-LRP is equivalent to Input $\times$ Gradient (IxG) under ReLU nonlinearities. In order to have a more comprehensive demonstration, we include more attribution methods such as LRP, guided GradCAM, gradient SHAP, etc. in the comparison. The results have been updated to the **appendix G**.
>
>
> **[Blurring Reference Implementation]**
>
> Let $\mathbf{x}\in\mathbb{R}^D$, the deletion of features is implemented as $x_{deleted}=\mathbf{x}\odot\mathbf{m}+\mathbf{r}\odot(1-\mathbf{m})$ where $\mathbf{r}\in\{0,1\}^d$ is the binary mask determined by the trajectory/heatmap/number of features to be deleted. And $\odot$ is the Hadarmard product that performs feature-wise deletion. $\mathbf{r}\in\mathbb{R}^d$ is the reference values. In practice, we change the reference values through the definitions of $\mathbf{r}$. That is, $\mathbf{r}\_{zero}=\mathbf{0}, \mathbf{r}\_{mean}=\bar{x}\mathbf{1}$. As for blurring, we use $\mathbf{r}\_{blur} = \texttt{Blurrer}(\mathbf{x})$, where $\texttt{Blurrer=torchvision.transforms.GaussianBlur(kernel\\_size=(31, 31), sigma=(56, 56))}$.
>
>
>
> **[Fig 2a Illustration]**
>
> We thank the reviewer for the suggestions! We would first want to elaborate that there are no -GO curves because it is unsolvable.
> In Figure 2a, we demonstrate how well the TRACE solved by simulated annealing can approximate the global optimum (-GO) by approaching the unreachable bound of complete search (CS). And the global optimum is hence squeezed between them, as the Cyan and Lime color regions. The fact that these two regions are too slim to recognize actually verifies that TRACE-SA is able to approximate the theoretical global optimum very closely compared with the large AUCs.
>
> **[Elaborations on Table 1]**
>
> We thank the reviewer for pointing out the ambiguity in the analysis of table 1.
>
> - Since the ratio $\frac{\textrm{variance of TRACE}}{\textrm{improvement from TRACE}}$ is too small, we do not repeat the experiments for the same (model, reference, input) tuple. Also, it should be noted that back-propagation-based explanation methods are also deterministic. As a result, **for the (model, reference) pair, we report the mean LeRF-MoRF score over samples.**
> - For the analysis of TRACE and the metric, the analysis of the results shown in table 1 can be found in sections 5.1, 5.2. Here we summarize them as follows:
>  (1) In each row, we verify the TRACE's optimality.
>  (2) In the TRACE-p and TRACE-y columns we demonstrate that the difference between "using probability" and "using logits" as the indicator is not negligible and should be decided carefully w.r.t. the evaluation purpose.
>  (3) Comparing the three blocks with different reference values, we conclude that with the selected settings, the differences caused by reference values are almost negligible.
>
> - For the analysis of existing explanation methods, we can also draw insightful conclusions. Please see the response in **[Rankings of Baselines]**.
>
> **[Details in Notions, References and Grammar]**
>
> We thank appreciate the reviewer's careful and detailed suggestions! We've revised the mentioned notions, typos and grammar mistakes.
>
>
> **References**
>
> [1] "Evaluating the Robustness of Interpretability Methods through Explanation Invariance and Equivariance", NeurIPS 2023
>
> [2] "On pixel-wise explanations for non-linear classifier decisions by layer-wise relevance propagation", PloS one, 10(7), e0130140.
>
> [3] "Towards better understanding of gradient-based attribution methods for deep neural networks", ICLR 2018

---

### Official Review · Reviewer_HLaZ · 2023-10-27

**Soundness:** 3 good
**Presentation:** 2 fair
**Contribution:** 2 fair
**Rating:** 3
**Confidence:** 4

**Summary:**

In this paper, the authors report the evaluation results of insertion/deletion metrics (MoRF, LeRF) across several benchmarks.
They proposed a method called TRACE to approximately optimize MoRF and LeRF using simulated annealing or greedy algorithms, and compare TRACE with existing saliency map methods using LeRF and MoRF.
As a result, TRACE-Mo, which optimizes MoRF, outperforms existing methods in terms of MoRF, and TRACE-Le, which optimizes LeRF, achieves superior results compared to existing methods in the context of LeRF.
The authors also conducted experiments varying the background used to fill the deleted pixels and by altering the sizes of the deleted pixels.
These experiments demonstrate that TRACE consistently yields favorable results irrespective of the background choice, and that larger sizes of deleted pixels tend to produce stable outcomes.
Moreover, while TRACE-Le performs well for both MoRF and LeRF, TRACE-Mo yields less favorable results for LeRF.
Consequently, the authors concluded that LeRF is a preferable evaluation metric for saliency maps based on these findings.

**Strengths:**

The strength of this paper is the in-depth investigation of TRACE-Mo and TRACE-Le, which optimize MoRF and LeRF, respectively.

**Originality, Quality**

The authors compared simulated annealing and greedy algorithms as optimization methods for TRACE-Mo and TRACE-Le.
Furthermore, they reported that these approximate solutions yield practical and favorable results by exploring the lower bounds of MoRF and LeRF.
These comparative studies are both innovative aspects of this research and essential evidence that reinforce the validity of the experimental results.

**Clarity, Significance**

Please refer to the "Weaknesses" below.

**Weaknesses:**

The weaknesses of this paper include "question about the validity of the research approach" and "the absence of important related studies."

**Question About the Validity of the Research Approach**

If I understand correctly, the primary aim of this research is to investigate the properties of MoRF and LeRF in an explanation-method-agnostic fashion.
The authors propose to achieve this by directly optimizing MoRF and LeRF.
However, there is a lack of description of how directly optimizing these metrics leads to an explanation-method-agnostic investigation.
The direct optimization of MoRF and LeRF, as proposed in this study, gives rise to explanation methods such as TRACE-Mo and TRACE-Le.
The use of TRACE-Mo and TRACE-Le could be considered explanation-method-dependent rather than agnostic.

**Absence of Important Related Studies**

While this research suggests the direct optimization of MoRF and LeRF, similar studies have been conducted in the past.
One notable early study is [Ref1], which proposed optimizing MoRF as a continuous relaxation problem using gradient-based methods.
It also reported that the optiimzation of MoRF tends to generate artifact noise.
[Ref2] also proposed optimizing MoRF and LeRF using greedy methods or continuous relaxation with gradient-based optimization.
It also reported adversarial noise appears in the context of MoRF optimization.
Furthermore, [Ref2] concluded that LeRF is a more reliable metric than MoRF, as observed in this research.
Many of the key results in this research have already been established in these related studies.

Additionally, there is a relevant study [Ref3] regarding saliency map evaluation metrics.
[Ref3] investigated and evaluated the properties of MoRF and LeRF using a different approach than both this research and [Ref2].
Given its distinct perspective on exploring MoRF and LeRF properties, [Ref3] can be considered an important related study.

* [Ref1] "Interpretable Explanations of Black Boxes by Meaningful Perturbation," ICCV, 2017.
* [Ref2] "Feature Attribution As Feature Selection," OpenReview, 2018.
* [Ref3] "Sanity Checks for Saliency Metrics," AAAI, 2020.

As a minor weakness, Theorem 3.1 contains an error.
MoRF and LeRF are defined as sums of f, as shown in Equation (1), while the proof in Theorem 3.1 defines them as sums of x.
This proof is only valid for the case where f is a linear function and is inappropriate for general f.

**Questions:**

* What is the new finding of this current paper over [Ref1] and [Ref2]?
* How is the optimization of MoRF or LeRF relevant to the investigation of their properties?
* Why the optimization of MoRF or LeRF can be considered as explanation-method-agnostic approach?

---

> ### Author Response · Authors · 2023-11-18
> **Response to Reviewer HLaZ**
>
> **[The Outline of the Response]**
>
> We thank the reviewer for the insightful comments and important references. Here we address the reviewer's two main concerns in the following manner:
>
> 1. **[The Validity and the Purpose of Optimization, and the Explanation-Method-Agnostic Claim of TRACE]** This point is connected to the First two questions brought up by the reviewer. Therefore, we address the posed questions by answering how the existing problems are resolved by TRACE.
>
> 2. **[Additional Related Work]** After addressing the concern regarding the validity, we carefully review the related work mentioned by the reviewer. We review [Ref3] first and then answer Q3 by reviewing [Ref1,2]. These important related works have also been cited properly in our updated manuscript.
>
>
>
> **[Q1: How is the optimization of MoRF or LeRF relevant to the investigation of their properties?]**
>
> 1. **The optimum of a metric illustrates what the metric expects from the candidates (explanation methods) being measured.**
>
> If the best- or very-high-performing explanation (principled explanation) under a certain metric does not make sense, it indicates such a metric is improper.
> We draw an analogy to evaluation metrics in classification. In highly imbalanced data, directly optimizing (i.e. searching) for a classifier that achieves very-high-performing accuracy score is likely to result in a model that blindly classifies all samples into the majority class. And thus we can verify that the accuracy metric is improper in this scenario.
>
>
>
> 2. **A Formalized Elaboration**
>
> We elaborate on this using the following formulation.
> Given a black-box prediction $f(\mathbf{x})$, a metric can be seen as a mapping $M_{f,x}:\Psi\rightarrow \mathbb{R}$, where $\Psi$ is the set of all explanations w.r.t. the prediction $f(\mathbf{x})$. In order to study whether the metric $M_{f,x}$ is legitimate, TRACE aims at finding the optimizer $\psi^* = \arg\max_{\psi\in\Psi}M_{f,x}(\psi)$ of the metric, i.e., the "principled explanation" that optimizes the deletion metrics. Such principled explanation $\psi^*$ intrinsically reviews the deletion metric and represents "what the metrics are expecting from explanations".
>
>
> 3. **The Problem of Studying the metric $M\_{f,x}$ without its optimizer $\psi^\*$**
>
> Existing studies of metrics highly rely on explanation **methods**. They inspect the metrics by checking *the scores obtained by well-accepted explanation methods*. For instance, [1,2] rely on popular methods such as IG, Gradient, etc. to investigate deletion metrics.
> This is equivalent to checking their scores $M_{f,x}(\phi_{IG}),M_{f,x}(\phi_{Gradient}),\cdots$, which can be problematic for the following reasons:
>
>   (1) Since deletion metric $M_{f,x}(\psi) = \sum_{k=0}^df(\mathbf{x}\_{\sigma(\psi)[:k]})$ (using MoRF as an example) is infeasible to study analytically, enumerating these explanation points $\phi_{\texttt{explanation}}\in\Psi$ does not help in exploring the property of the metric $M_{f,x}$ because there's no way to tell if their scores are high or low.
>
>   (2) Since $M_{f,x}$ is discontinuous, even the relative score $M\_{f,x}(\psi\_{\texttt{explanation1}})-M\_{f,x}(\psi\_{\texttt{explanation2}})$ between two explanations provide no insights in understanding the metric.
>
> As a result, the optimizer $\psi^*$ of $M_{f,x}$ solved by TRACE might be the only property of the metric that we can have access to. And we study the deletion metric $M_{f,x}$ through this optimizer to investigate whether the metric $M_{f,x}$ is legitimate itself.
>
>
> 4. **An Example of the Use of the Optimizers**
>
> By revealing the principled explanation, we explore the connection/conflict between different metrics, by answering the question **"Does the best-performing explanation under one metric work well in other metrics"?** For example, in Section 5.2, we studied the connection/conflict between different deletion criteria, MoRF and LeRF, resulting in two metrics $M^{MoRF}\_{f,x}$ and $M^{LeRF}\_{f,x}$ with optimizers $\psi^{MoRF}, \psi^{LeRF}$. We thus employ $M^{LeRF}\_{f,x}(\psi^{MoRF})$, $M^{MoRF}\_{f,x}(\psi^{LeRF})$ to inspect whether these metrics are legitimate.
>
> Here, we draw an analogy to another important topic in trustworthy AI: fairness, where the connection/conflict among fairness notions has been widely studied [3,4]. There is an intrinsic conflict between two common group fairness metrics: demographic parity (DP) and equalized odds (EOd), since a classifier with the best EOd result may not ensure the best DP result. Also, a classifier with the best accuracy may not ensure the best DP result. A similar discussion on the connection/conflict among evaluation metrics in XAI is crucial yet underexplored, partly due to the challenge of identifying optimal explanations. TRACE enables us to systematically investigate this aspect.

---

> ### Author Response · Authors · 2023-11-18
> **Response to Reviewer HLaZ**
>
> **[Q2: Why the optimization of MoRF or LeRF can be considered as explanation-method-agnostic approach?]**
>
> We would like to clarify that TRACE is considered as an "explanation-method-agnostic" study of the deletion metric $M_{f,x}$ because it investigates the metric through its optimizer $\psi^*$, differing from existing studies that rely on attribution explanations $\psi_{IG},\psi_{Gradient},\psi_{GradCAM},\cdots$. Specifically, these explanations are independent of the metric $M_{f,x}$, and using $M_{f,x}(\psi_{IG}),M_{f,x}(\psi_{Gradient}),M_{f,x}(\psi_{GradCAM})$ to evaluate the metric itself can be greatly biased by these explanations. Oppositely, the optimizer $\psi^*$ is determined solely by the metric $M_{f,x}$. Using $M_{f,x}(\psi^*) = max_{\Psi}M_{f,x}$ as the pipeline to evaluate the metric is not biased by any existing explanation methods, and thus is considered explanation-method-agnostic.
>
> On the other hand, we would clarify that although the optimizer is referred to as "the principled explanation", it is completely derivative from the metric itself, and should not be confused with existing explanations.
>
>
> **[The studies of Deletion Metrics in [Ref3]]**
>
> The deletion metric is referred to as the Area Over the Perturbation Curve (AOPC) in [Ref3]. The work explores the AOPC scores of multiple attribution explanation methods such as SHAP, IxG, etc., and under MoRF/LeRF, respectively. [Ref3] fall short in a similar manner as [1,2] to rely on these explanation methods. Without the principled explanation that illustrates what the metric is really measuring, [Ref3] mainly serve as a sanity check for the deletion metric, as they point out the fact that "MoRF and leRF are highly-variable across saliency methods" but do not provide mitigations for the issue.
>
>
> **[Q3: New Findings Comparing with Related Work [Ref1] \& [Ref2]]**
>
> First, we would emphasize that **both [Ref1] and [Ref2] do not optimize the deletion metric (i.e. MoRF, LeRF, LeRF-MoRF), nor do they optimize any smooth or relaxed version of the deletion metric.** We now review the two references and compare our work with them in details. Note that the proposed two perturbation-based methods share many common characters, so we begin by discussing the shared differences. Then we delve into the differences between them and TRACE separately.
>
> 1. **General Review of [Ref1, Ref2]**
>
> [Ref1] and [Ref2] both aim at generating relaxed masks so that the prediction of the masked input is minimized and are perturbation-based explanation methods. Both of them use a single continuous mask $\mathbf{m}\in[0,1]^d$ to perturbe the input image through convex combination $\bar{\mathbf{x}}_{m,r} = (\mathbf{1}-\mathbf{m})\odot\mathbf{x} + \mathbf{m}\odot\mathbf{r}$. And they aim at solving
>
> - [Ref1]: $\mathbf{m}^*=\arg\min_{[0,1]^d}\lambda\|\mathbf{1}-\mathbf{m}\|\_1+f(\bar{\mathbf{x}}_{w,r})$. (eq (3) of [Ref1])
> - [Ref2]: $\mathbf{m}^*=\arg\min_{[0,1]^d}\sum_{i=1}^d(1-m_i)+\lambda\mathbb{E}[f(\bar{\mathbf{x}}_{w,r})]$. (eq (3.2) of [Ref2])
>
> *(In order to universally review [Ref1,Ref2], we unify the notations of the two work.)*
>
> It can be observed that the only difference is that [Ref1] use pre-defined references while [Ref2] use the expectation over random noise references.
> On the other hand, TRACE, aiming at optimizing the deletion metric, is defined as follows:
>
> - TRACE-Mo: $\tau^*=\arg\min_{\tau}\sum_{k=0}^df(\mathbf{x}_{\tau[:k]})$
>
> We can observe that there's almost **no** similarity between TRACE's objective (the deletion metric) and [Ref1,Ref2]. e.g.,
>
> (1) [Ref1,2] solve for a one-time deletion mask that minimize the model's prediction once applied, but TRACE follows the deletion test to include all possible number of deleted features to optimize the progressive deletion process.
>
> (2) Therefore, [Ref1,2] are only able to guarantee the performance of feature deletions with a fixed number of deleted features, while TRACE guarantees all.
>
> (3) In fact, [Ref1,2] are more similar to the smooth/relaxed version of a *single term* of the complete search (CS) in our manuscript. In complete search, a single term can be seen as a minimization with $L_0$ norm on the mask (fixed number of features perturbed).
>
> (4) The relaxation enables differentiable optimization, but even for their own objective, the optimality has not been rigorously studied, indicating that they might fall into local optimum and there are better masksl, let alone the deletion metric. As a principled explanation, TRACE's optimality for the deletion metric has been verified through the squeezing between the complete search.
>
> (5) The relaxation of the masks make the explanations proposed by [Ref1,2] intrinsically incompatible with the deletion metric, where features are deleted with strict binary masks in each term.
>
> etc.
>
> In summary, the only common point might be they all aim at finding a way to minimize/maximize the models prediction. But that holds true for all perturbation-based methods.

---

> ### Author Response · Authors · 2023-11-18
> **Response to Reviewer HLaZ**
>
> **[Q3: New Findings Comparing with Related Work [Ref1] \& [Ref2]]**
>
> 2. **Details of [Ref1]**
>
> [Ref1] indeed point out that deleting features to minimize the models' prediction can create artifacts, but [Ref1] mitigate this by introducing stochasticity in $m$ and enforcing a total-variation normalization. Although they are effective in generating the perturbation-based attribution method, they provide no help in mitigating the OOD problem in the deletion metric whatsoever.
>
> TRACE, on the other hand, is developed to benchmark the most popular and widely used deletion metric. Thus almost all the conclusions regarding the metric that are drawn from our work by TRACE differ from [Ref1].
> For instance, even for the artifacts discussed in both works, TRACE suggests that they are inevitable in the progressive deletion with small patch sizes, but using LeRF$-$MoRF or LeRF instead can mitigate the issue greatly. [Ref1], though describing the "preservation game" very briefly in section 4.2, do not provide any further analysis of its effectiveness.
>
>
> 3. **Details of [Ref2]**
>
> [Ref2], although similar to [Ref1] as a perturbation-base method, do carry out experiments testing the proposed methods with the deletion metric and verify that [Ref2] outperform the selected explanation methods. However, it falls short in the following aspects.
>
> (1) **The optimality is important for the principled explanations to benchmark the metrics.**
>
> Although [Ref2] outperform the selected attribution methods in the deletion test, there's no guarantee for the optimality theoretically (e.g. optimization objective) or empirically (e.g. the comparison between CS to quantify -GO). Therefore, it only demonstrates that "[Ref2] are better than **these selected explanation methods**", or formally $M_{f,x}(\psi_{\texttt{[Ref2]}}) > M_{f,x}(\psi_\texttt{selectedExplanations})$. Recall that in the previous analysis of Q1, point 3, the relative difference of scores does not have any insights due to the discontinuity of $M_{f,x}$. There's no guarantee whether the next selected explanation methods will outperform them.
>
> Besides, it is noticed that the selected methods are those that are verified to be inconsistent with the deletion metrics in our manuscript. Differently, TRACE is guaranteed to outperform **all attribution explanation methods** in the deletion metric. This enables us to perform rigorous studies of the deletion metric by checking the TRACE results.
>
> (2) **Both Directions Matter**
>
> By combining LeRF and MoRF, we conclude that although LeRF is much better than MoRF and is less affected by the OOD issue, LeRF$-$MoRF is actually the best criterion for the deletion metric, and outperforms LeRF by a large margin. The corresponding TRACE-Le$-$Mo provides the principled explanation. However, from the optimization objective, it can be noticed that [Ref2] can not solved for combined directions.
>
> (3) **Studies Focusing on the Deletion Metrics**
>
> Using the optimizer $\psi^*$, we carry out a rigorous study over the question "What are the metrics really expecting from the explanations?", and propose many practical guidelines on how the deletion metrics should be used and how the variants are selected. These regulations on how to evaluate explanation methods are important but missing from these references.
>
>
>
>
>
>
> **[Typo of Proof of Theorem 3.1]**
>
> We thank the reviewer for pointing this typo. In the proof of theorem 3.1, all the terms in eq (5) should be $f(x)$ instead of just $x$. In fact, given a trajectory $\tau$, for *deletion with MoRF*, the AUC is written as $f(x_{\backslash\tau[d:]}) + f(x_{\backslash\tau[d-1:]})+\cdots+f(x_{\backslash\tau[0:]}) = \sum_{k=0}^df(x_{\backslash\tau[k:]})$. Similarly, for *insertion with LeRF*, the AUC is written as $f(x_{\tau[:0]})+f(x_{\tau[:1]})+\cdots+f(x_{\tau[:d]}) = \sum_{k=0}^df(x_{\tau[:k]})$. Since deleting the least important $k$ features is equivalent to inserting the most important $d-k$ features, we have $\sum_{k=0}^df(x_{\backslash\tau[k:]})=\sum_{k=0}^df(x_{\tau[:k]})$.
>
>
> **References**
>
> [1] "A Benchmark for Interpretability Methods in Deep Neural Networks", NeurIPS 2019
>
> [2] "A consistent and efficient evaluation strategy for attribution methods", ICML 2022
>
> [3] "FACT: A diagnostic for group fairness trade-offs", ICML 2020
>
> [4] "Translation tutorial: 21 fairness definitions and their politics", Proc. conf. fairness accountability transp., new york, usa. Vol. 1170. 2018.
>
> [Ref1] "Interpretable Explanations of Black Boxes by Meaningful Perturbation", ICCV 2017
>
> [Ref2] "Feature Attribution As Feature Selection", Openreview 2018
>
> [Ref3] "Sanity Checks for Saliency Metrics"

---

> > ### Comment · Reviewer_HLaZ · 2023-11-23
> > **Re: Response to Reviewer HLaZ**
> >
> > First, I would like to thank the author for the detailed discussions on the points I raised.
> > I think I now better understand the conceptual as well as methodological differences.
> >
> > However, I recently came across with a very similar paper.
> > * The Solvability of Interpretability Evaluation Metrics, ECML 2023 (Findings)
> > * https://yilunzhou.github.io/solvability
> >
> > This paper also assesses the metrics by directly optimizing them.
> > Their comprehensiveness $\kappa$ and sufficiency $\sigma$ seems to be identical with the TRACE objectives.

---

> ### Author Response · Authors · 2023-11-23
> **Response to Reviewer HLaZ**
>
> We thank the reviewer very much for pointing out this very recent related work. We will address the reviewer's concern by explaining the difference between our work and [Ref3].
>
> 1. **The optimality is still not guaranteed in [Ref3].**
>
> As we've agreed that optimality is essential for the principled explanation of a metric, we first discuss how [Ref3] compromise in achieving the optimality.
>
> (1) As shown in Algorithm 1 of [Ref3], beam search is used to search for a solution. In each iteration, beam search extends to find the next feature to add to the solution $e^*$. Such a step-by-step search is in fact a relaxed greedy scheme that does not take the entire trajectory into consideration. Therefore, it suffers from all the drawbacks of the standard greedy search that simply searches for the next feature to insert/delete to maximize/minimize the prediction of the next step.
>
>
> (2) The justification for the use of beam search is technically deprecated.
>
> It is mentioned in [Ref3] that the beam search is effective in optimizing the objective "because it's 'additive'" on Page 5, left column, under Algorithm 1. The "additive property" is defined in Definition 4.1 that if a metric $m(x,e)$ (where $e$ is the explanation under their notations) can be written as a summation $m(x,e)\sum_{l=1}^L h(x, e^{(l)})$ for sum $h$, where "$e^{(l)}$ reveals the attribution values of $l$ most important features according to $e$ but keeps the rest inaccessible."
>
> It should be noted that these arguments are technically flawed because $e^{(l)}$ and $e^{(l+1)}$ are never independent, let alone additivity. On the contrary, they are actually incremental and highly correlated. Hence greedily searching for the next step does not optimize the entire targeted AUCs.
>
> (3) Since the optimality is not supported from the theoretical perspective, [Ref3] is essentially equivalent to [Ref2] in *finding an explanation method that outperforms the selected existing explanation methods under the deletion metric, but the optimality is never guaranteed.* As a result, [Ref3] is not the principled explanation. Without the optimality, [Ref3] suffers from the exactly same issue as [Ref2] as shown in point 3 of the review of [Ref2]:
>
>   - Since deletion metric $M\_{f,x}(\psi) = \sum\_{k=0}^df(\mathbf{x}\_{\sigma(\psi)[:k]})$ (using MoRF as an example) is infeasible to study analytically, enumerating these explanation points $\phi_{\texttt{explanation}}\in\Psi$ does not help in exploring the property of the metric $M_{f,x}$ because there's no way to tell if their scores are high or low.
>
>   - Since $M_{f,x}$ is discontinuous, even the relative score $|M_{f,x}(\psi_{\texttt{explanation1}})-M_{f,x}(\psi_{\texttt{explanation2}})|$ between two explanations provide no insights in understanding the metric.
>
>
> (3) As a greedy algorithm, beam search is also short-sighted and it thereby cannot include both directions (MoRF and LeRF) together. As we've verified in our work LeRF is only a compromise to LeRF$-$MoRF.
>
> (4) In contrast, TRACE treats the solution trajectory as a whole and makes use of combinatorial optimization schemes so that all the issues above are resolved. For example, (a) Instead of just the several selected baseline explanation methods, TRACE is guaranteed to outperform **all attribution explanation methods** in the deletion metric. (b) Solving for the entire trajectory $\tau$ enables us to optimize the objective LeRF$-$MoRF directly. (c) Also, the true optimality of TRACE-SA is demonstrated by comparing it with the theoretical global optimum (TRACE-GO) and Complete Search (CS). etc.
>
>
> 2. **Benchmarking the Deletion Metrics**
>
> In conclusion, [Ref3] aims to develop an explanation method that achieves better scores under the deletion metric by searching the feature to delete step-by-step. The discussion is also mainly focused on using it as an explanation while the objective, the deletion metric itself is not studied.
>
> Differently, TRACE is proposed to benchmark all the variants of deletion metrics to resolve the potential problems of the metric such as the infamous OOD issue, so that this favorable metric of attribution explanations can be used carefully and normatively. As we have agreed, rigorous studies and analyses of this nature should be conducted using principled explanations $\psi^*$, a domain that is only achievable with TRACE and has not been explored previously. The application of such a theoretically and empirically validated optimal trajectory as a method for deletion-wise explanation not only fulfills this criterion but also adds significant value to the field.
>
>
> **References**
>
> [Ref2] "Feature Attribution As Feature Selection", Openreview 2018
>
> [Ref3] "The Solvability of Interpretability Evaluation Metrics", EACL 2023 (Findings)

---

### Official Review · Reviewer_VCEJ · 2023-11-01

**Soundness:** 2 fair
**Presentation:** 2 fair
**Contribution:** 2 fair
**Rating:** 5
**Confidence:** 3

**Summary:**

The work presents a framework for evaluating deletion metrics in explanation techniques for vision tasks. The approach proposes a formalization of the problem that relies on combinatorial optimization for finding good deletion trajectories and is evaluated on Imagenet and ResNet-18 (and other convolutional models).

**Strengths:**

S1 - The formalization of the TRACE framework is meaningful and can be a useful tool for comparing different types of explanations.

S2- The framework also has considerations for the OOD problem in explanations, which arise when features are replaced with "null or zero" values during deletion leading therefore to unnatural images.

**Weaknesses:**

W1- The paper has major presentation and linguistic problems in several areas. First, the main body of the paper does not provide enough intuition or sketches for the theoretical proofs. At the very least, the main body of the paper should provide a sketch of the proof flow. Second, the related work is moved to the appendix, and making it difficult for the reader to understand or position the novelty of the work with respect to previous work. Third, in several places, the paper is written too vaguely or in an informal manner, which makes it difficult to judge the specifics of the results. In particular, the introduction, the discussion of results and conclusions miss important specifics or at least linguistically difficult to understand. Lastly, the paper requires a revision throughout to make sure that the notation is consistent within the main body of the paper and the appendix, and that all explanation methods and acronyms are properly defined and introduced.

W2- The paper does not take a strong position on making claims about which explanation methods perform better according to the TRACE framework. Results are mostly presented numerically but they are not discussed analytically.

**Questions:**

Further questions:

- How does TRACE deal with correlated features?

- How does (or may) TRACE operate on tabular datasets?

- Do the authors observe any qualitative differences between the different model architectures and size when it comes to the quality of the generated explanations?

---

> ### Author Response · Authors · 2023-11-18
> **Response to Reviewer VCEj**
>
> We thank the reviewer for the suggestions and questions. We will address all of them in detail.
>
> **[W1: Presentations]**
>
> We address the reviewer's concern point-by-point.
>
> 1. **[Intuitions on Proofs]**
>
>   Our work presents three main proofs. They demonstrate the following facts:
>
>   - B.2/B.3: We show that the attribution maps can be simplified to trajectories with mathematical rigors. This is done by showing that the trajectories can be seen as equivalence classes of attributions. The intuition behind this is clearly stated in *section 2. Trajectory Importance (TRACE), page 3* of the manuscript.
>
>   - 3.1: We show that we distinguish between MoRF and LeRF, then there's no need to distinguish between deletion and insertion anymore, because deletion-MoRF is equivalent to insertion-LeRF, and deletion-LeRF is equivalent to insertion-MoRF. The intentions and intuitions are clearly discussed in *section 3, page 4, the Deletion vs. Insertion paragraph*.
>
>   - 4.1: We show that the optimization problem for TRACE is NP-hard. This is to justify the use of combinatorial optimization tools for solving the optimization problem. The intuitions for the proof are discussed in *the first paragraph of section 4*.
>
> We believe both the statements and the proofs are clear to follow. We would appreciate it if the reviewer could point out the part where the proofs may appear obscure or misleading so that we can improve the presentation of the proofs accordingly.
>
> 2. **[Related Work]**
>
> We acknowledge the reviewer's concern regarding the placement of related work.
> We would clarify that **all** the important related work (i.e. the studies of the explanation metrics and tested attribution methods) are thoroughly discussed in the introduction section (paragraphs 2-4) of the manuscript. This choice was made deliberately to enhance the reading flow and ensure that essential background information is immediately accessible to the reader.
>
> We assure that the additional related work section in the appendix is not crucial for understanding any arguments and findings of our study. Instead, it serves as an extensive reference list for those interested in a deeper exploration of XAI studies focused on attribution explanations.
>
> It contains two parts, the attribution methods and attribution metrics. The first part elaborates on the wide variety of existing attribution methods, including those we have referenced previously for our experiments. The second part provides an overview of the research landscape concerning attribution metrics, essentially offering a different perspective to the introduction. It transitions from a point-guided manner to a work-guided manner.
>
>
> 3. **[Concerns Regarding the Clarity]**
>
> We thank the reviewer for their valuable feedback regarding the clarity and formality of certain sections of our manuscript.
> We take these comments seriously and are committed to enhancing our manuscript where "important specifics" are missing or the contexts are "linguistically difficult" to understand. We'd appreciate it if the reviewer could point out these areas, and we will make targeted revisions to address these concerns.
>
>
> 4. **[Notions & Acronyms]**
>
> We appreciate the reviewer for mentioning the problems in the notions and acronyms. We've carried out a revision of the manuscript to ensure the notions and acronyms are consistent.

---

> ### Author Response · Authors · 2023-11-18
> **Response to Reviewer VCEj**
>
> **[W2: Claims over Existing Explanation Methods]**
>
> The proposed TRACE framework aims at benchmarking the popular deletion metric for explanation methods using the principled explanation -- **without relying on any existing explanation methods**. Thus the analysis and the conclusions were not focused on analyzing specific existing explanation methods.
> But with the benchmarked variants of the deletion metrics, we sure can have the following rankings of existing explanations under the verified deletion metric.
>
> (1) The ranking illustrates "*to what extent can the selected features of these explanation methods affect the model's behavior*":
> - Probability: GradCAM (GC) $>$ Information Bottleneck Attribution (IBA) $>$ RISE $>$ Excitation Back-Propagation (EBP) $>$ RISE $>$ Extremal Perturbation (EP) $>$ Gradient (Grad) $>$ Integrated Gradient (IG) $>$ Input $\times$ Gradient (IxG)
> - Logits: GradCAM (GC) $\approx$ RISE $>$ Information Bottleneck Attribution (IBA) $>$ Excitation Back-Propatation (EBP) $>$ Extremal Perturbation (EP) $>$ Gradient (Grad) $>$ Integrated Gradient (IG) $>$ Input $\times$ Gradient (IxG)
>
> The results suggest that (a) GradCAM is the state-of-the-art among all existing attribution methods for selecting important features that maintain the model's performance when kept and harm the model's performance when deleted; and (b)perturbation-based methods (eg. RISE, EP, etc.), although related more closely to feature perturbations, do not have significant superiority over back-propagation-based method (GradCAM, EBP, Gradient, etc.) under the deletion metric. In fact, (3) it is noticed that pixel-wise attributions (e.g. IG, IxG and Gradient) tend to underperform the deletion metrics. It is the feature size (pixel-level, patch-level, etc.) that affect more in the deletion metric rather than the difference between back-propagation and perturbation.
>
> (2) Therefore, we advocate that the metrics for explanations should serve specific and practical purposes. For instance, deletion metrics measure "to what extent can the selected features of these explanation methods affect the model's behavior", Invariance/Equivariance Robustness [1] measures whether the highlighted features are invariant/equivariant w.r.t. transformations, etc.
>
> (3) By finding the optimal explanation of the deletion metrics, we also emphasize that the evaluation of existing explanation methods should not be based on a single and universal metric. Failing certain metrics does not instantly disprove explanation methods (e.g. gradient in deletion test). It's also likely that the metrics and the explanation methods focus on different goals.

---

> ### Author Response · Authors · 2023-11-18
> **Response to Reviewer VCEj**
>
> **[Correlated Fatures]**
>
> Due to the highly non-linearity of DNNs, features are usually correlated in the forward pass. However, this nonlinearity makes it impossible to study the behaviors of models. As a result, locally linear/additive/decoupling assumptions are usually made in the studies of the mechanism of DNNs, especially in attribution methods.
> As such, the deletion/insertion metric itself makes this compromise, too, where features are deleted/inserted independently. That is, given an attribution map $\phi_f(x)\in R^d$, even if the ith and the jth features are correlated, the input features are still perturbed independently based solely on the relation between $\phi_f(x)_i$ and $\phi_f(x)_j$.
> The correlated features, when deleted independently, can lead to confusion, because the order of their deletions matters.
> **Fortunately, TRACE takes *all* possible deletion trajectories into consideration and solves for the optimizer trajectory, which mitigates the influence of the ordering of correlated features.**
>
> We present an example of correlated features. In order for the ground truth to be known and controllable, suppose $f(x)=x_1+ax_2+x_3+2x_1x_2, x=(1,1,1)$, where $x_1,x_2$ are correlated. Given a trajectory $\tau$ of the 3 features, let $s(\tau)$ be the corresponding LeRF$-$MoRF scores under the deletion metric. A higher score indicates a better trajectory under the deletion metric.
>
> | $a$      | (1,2,3) | (1,3,2) | (2,1,3) | (2,3,1) | (3,1,2) | (3,2,1) |
> | ----------- | ----------- | ----------- | ----------- | ----------- | ----------- | ----------- |
> | 0.5   | -2.0 | -1.0 | -1.0 | 1.0 | 1.0 | **2.0** |
> | 0.9   | -2.0 | -0.2 | -1.8 | 0.2 | 1.8 | **2.0** |
> | 1.0   | -2.0 | 0.0 | -2.0 | 0.0 | **2.0** | **2.0** |
> | 1.1   | -2.0 | 0.2 | -2.2 | -0.2 | **2.2** | 2.0 |
> | 2.0   | -2.0 | 2.0 | -4.0 | -2.0 | **4.0** | 2.0 |
>
>  We can see that using TRACE, the trajectories are found in the following ways:
>
>   (i) when $a=1$, $x_1,x_2$ are symmetric. The LeRF-MoRF scores for the 6 trajectories are shown in the third row. Both $\tau=[3,1,2]$ and $\tau=[3,2,1]$ are optimal, suggesting that $x_1,x_2$ are equivalently important in the deletion metric.
>
>   (ii) when $a< 1$, $x_1$ is more important. In such case, we have in the top two rows that $\tau=[3,2,1]$ has the highest score. This verifies that TRACE recognizes $x_1$ as the most important one (last to delete).
>
>   (iii) when $a>1$, similarly, $x_2$ is more important. And from the last two rows, $\tau=[3,1,2]$ has the highest score. This verifies that TRACE recognizes $x_2$ as the most important one, too.
>
>   **This example shows that based on the prediction change w.r.t. the deletion metric, TRACE is able to detect the relative importance of the correlated features to the model recognized by the deletion metrics.**
>
> Additionally, for image datasets, it should be noted it's a general consensus that pixels that are spatially close usually tend to have higher correlations. Therefore, as TRACE suggested, casting deletions/insertions w.r.t. the superpixel patches instead of the small pixels individually, is already an approach to take the correlation of features into consideration.
>
>
>
>
> **[Tabular Datasets]**
>
> Based on perturbing input features, the optimality of TRACE is independent from the data modality, making it readily applicable to other domains. We've added experimental results on tabular data in **appendix G**, along with more detailed analysis. The results are consistent with image data.
>
>
> **[Insights of model architectures and sizes discovered by TRACE]**
>
> In fact, we can observe quantitative insights between model structures and model sizes. We've added experiments in **appendix H** to demonstrate how TRACE transfers
>
> Since TRACE trajectories illustrate the intrinsic influence of the feature deletion towards the models performance, the similarities between model structures and sizes can be quantitatively through the **transferability** of the trajectories.
> That is, "Will the model $f_2$ achieve the same level of AUCs following the *optimal* trajectory from model $f_1$?" The results have been added to the appendix, where we've verified the similarities between different sizes from same model families (e.g. mobilenet, resnet) and between different model families.
>
>
>
>
> **References**
>
> [1] "Evaluating the Robustness of Interpretability Methods through Explanation Invariance and Equivariance", NeurIPS 2023

---

> > ### Comment · Reviewer_VCEJ · 2023-11-22
> > **Post rebuttal**
> >
> > I'd like to thank the authors for their detailed answer! The answer does address quite a few of the issues I had raised initially, in particular regarding the open questions around the empirical results and correlated features.
> >
> > I have adjusted the score accordingly. However, the work still requires improvement when it comes to presentation and situation with related work. I definitely understand that the authors aimed at increasing readability and moving related work later, but the practice is questionable when it comes to scientific publications. Other reviewers have raised further relevant questions on related work.
> >
> > Presentation-wise, I'd suggest that the authors pay attention to the flow and major writing issues. For example, looking at page 6, the second paragraph is completely disconnected and starts with a typo, some of the new figures use the term probiction instead of prediction? as a y axis label, the last two paragraphs of conclusions require more concrete rewriting etc. Notation has improved but there are still symbols that are used to note two different things.

---

> > > ### Author Response · Authors · 2023-11-22
> > > **Response to Reviewer VCEj**
> > >
> > > We appreciate the reviewer greatly for the suggestions and acknowledgment of our work!  In response to the suggestions, we've further addressed the reviewer's concern as follows:
> > >
> > > - For the presentation of the related work, we agree with the suggestion that certain formats of scientific publications are preferred. In response to this,  we have restructured the first section on page 2  to provide a specific review of the related work in the studies of deletion metrics in XAI in the main manuscript.
> > >
> > > - For the potential notation issue, we deduce that the reviewer refers to the abbreviations of MoRF, TRACE-Mo, etc. Here we give an overall clarification on how they are distinguished. (1) When MoRF/LeRF/MoRF$-$LeRF are used alone, they refer to the specific settings for the deletion metric. (2) TRACE-Mo/TRACE-Le/TRACE-Mo$-$Le refers to the corresponding principled deletion trajectories solved by TRACE. They are solved using simulated annealing if not specified otherwise. (3) In the discussion of the optimality of TRACE, we include an additional notation to denote the optimization schemes.
> > > For example, TRACE-SA-Mo refers to the principled deletion trajectories of MoRF that are solved by simulated annealing, while TRACE-Greedy-Mo refers to those that are solved by the Greedy scheme. Besides, CS-Mo refers to the Complete Search results of the MoRF test, where there's no need to specify the optimization tool. Finally, TRACE-GO-Mo refers to the theoretical global optimizer of MoRF, which is bounded between TRACE-SA-Mo and CS-Mo.
> > >
> > > - For the typo and disconnected paragraph 2 on page 6, we wonder if the reviewer refers to the paragraph under the figure caption that starts with "gorithms,...". This is the continued paragraph where "al-" of the starting word "algorithms" can be found at the bottom of page 5.
> > >
> > > - We have also revised the labels of the new figure in the appendix for the typo on the y-axis.

---

### Official Review · Reviewer_8p78 · 2023-11-01

**Soundness:** 3 good
**Presentation:** 3 good
**Contribution:** 3 good
**Rating:** 8
**Confidence:** 3

**Summary:**

The authors propose a method for finding the best-performing deletion sequence under a given deletion metric as a way of comparing different deletion metric settings and existing attribution-based explanation methods. The authors frame this problem as a combinatorial problem and find a computationally feasible way of finding a near-optimal solution. They compare different approaches in this regard. They consider various settings for the deletion metrics and study the performance of their method experimentally on image data. They demonstrate that indeed their method finds the best-performing explanations vs existing explanation methods.

**Strengths:**

Deletion metrics are often used in XAI and should be better understood, so this work contributes to an important area.

The paper is well organised and easy to read.

**Weaknesses:**

The proposed method seems to focus mainly on understanding better the behaviour of deletion metrics themselves. The authors show that various explanation methods then have a significant gap to the optimal deletion-sequence generated by their method. It’s not clear to me if and how this insight could be used to also understand better how good a given explanation is.

**Questions:**

Nothing to add

---

> ### Author Response · Authors · 2023-11-18
> **Response to Reviewer 8p78**
>
> We thank the reviewer very much for the appreciation of our work! Here we address the reviewer's further concerns as follows.
>
> **[The insights for given explanation methods]**
>
> For an explanation generated by a given explanation method, we can draw the following conclusions:
>
> (1) With the benchmarked variants of the deletion metrics, we can have the following rankings of existing explanations under the verified deletion metric. The rankings illustrate "*to what extend can the selected features of these explanation methods affect the model's behavior*":
> - Probability: GradCAM (GC) $>$ Information Bottleneck Attribution (IBA) $>$ RISE $>$ Excitation Back-Propagation (EBP) $>$ RISE $>$ Extremal Perturbation (EP) $>$ Gradient (Grad) $>$ Integrated Gradient (IG) $>$ Input $\times$ Gradient (IxG)
> - Logits: GradCAM (GC) $\approx$ RISE $>$ Information Bottleneck Attribution (IBA) $>$ Excitation Back-Propatation (EBP) $>$ Extremal Perturbation (EP) $>$ Gradient (Grad) $>$ Integrated Gradient (IG) $>$ Input $\times$ Gradient (IxG)
>
> The results suggest that (a) GradCAM is the state-of-the-art among all existing attribution methods for selecting important features that maintain the model's performance when kept and harm the model's performance when deleted; and (b) perturbation-based methods (eg. RISE, EP, etc.), although related more closely to feature perturbations, do not have significant superiority over back-propagation-based method (GradCAM, EBP, Gradient, etc.) under the deletion metric. In fact, (c) it is noticed that pixel-wise attributions (e.g. IG, IxG and Gradient) tend to underperform the deletion metrics. It is the feature size (pixel-level, patch-level, etc.) that affect more in the deletion metric rather than the difference between the back-propagation and perturbation.
>
> (2) Therefore, we advocate that the metrics for explanations should serve specific and practical purposes. For instance, deletion metrics measure "to what extend can the selected features of these explanation methods affect the model's behavior", Invariance/Equivariance Robustness [1] measures whether the highlighted features are invariant/equivariant w.r.t. transformations, etc.
>
> (3) By finding the optimal explanation of the deletion metrics, we also emphasize that the evaluation of existing explanation methods should not be based on a single and universal metric. Failing certain metrics does not instantly disprove explanation methods (e.g. gradient in deletion test). It's also likely that the metrics and the explanation methods simply focus on different goals.
>
> The proposed TRACE framework aims at benchmarking the most popular deletion metric for explanation methods rather than the explanation methods themselves, and thus the analysis and the conclusions were not focusing on any specific existing explanation methods. However, we believe the insights mentioned above are important to the XAI studies.
>
> **References**
>
> [1] "Evaluating the Robustness of Interpretability Methods through Explanation Invariance and Equivariance", NeurIPS 2023

---

### Author Response · Authors · 2023-11-21
**We are happy to answer more questions if there still exist concerns for our paper.**

Dear Reviewers,

Thanks very much for the time and effort that you have dedicated to reviewing our paper. We greatly appreciate your constructive comments and hope that our response adequately addresses your concerns.

Should you have any further questions or confusion, we are more than willing to provide additional clarifications. Thank you again for your valuable insights.

Best,

Authors

---

### Meta-Review · Area_Chair_H4Rk · 2023-12-12

**Metareview:**

This work studies how to reliably evaluate attribution-based explanations method by inserting or deleting features. The authors frame this task as a combinatorial optimization problem where the goal is to generate trajectories. Given this problem, they propose a method -- TRACE -- that can generate near-optimal trajectories for incremental evaluation. The authors evaluate their method and its variants in experiments on real-world datasets, showing that it can produce improved deletion trajectories on image classification tasks in a way that maintains the plausibility of deletion.

*Strengths*

- Originality: Identifies a new problem in the evaluation of attribution-based explanation methods.

*Weaknesses*

- Significance: The paper fails to make a compelling case to improve the incremental evaluation of attribution-based explanation models. Some of this stems from the lack of evidence for how improving evaluation would lead to meaningful **downstream** benefits in model development. Put simply -- why should practitioners use the trajectories used by TRACE, rather than the trajectories used by other methods? How would decisions in model development change as a result of using attribution-based explanations from TRACE as opposed to the baselines? As it stands, the motivation for a principled approach boils down to statements such as:
- "such influential metrics for explanation methods should be handled with great scrutiny,"
- "metrics should undergo rigorous studies before widespread adoption,"
- "the metric’s widespread use suggests that such property is valued in the XAI community."

- Clarity (minor): Several reviewers noted that the clarity of the paper could be improved considerably. I agree with these comments but note that this is something that could have been easy to address in a potential revision.

*What's Missing*

I would recommend including material to highlight the practical benefits of the proposed method and the potential benefits/risks of proper/improper evaluation. This could be achieved by adding demonstrations to the current experimental section or in a standalone demonstration section where the authors put explanations "in action." Ideally, the material would show how the method would work on a concrete application and highlight some of the downstream risks of using suboptimal trajectories.

**Justification For Why Not Higher Score:**

The reviews and discussion all point to the fact that the paper needs to make a stronger case as to the importance of this problem and the significance of its solution. As it stands, this will require substantial changes and should be validated through another round of peer review.

**Justification For Why Not Lower Score:**

N/A

---

### Decision · Program_Chairs · 2024-01-16

Reject